# Comparative analysis of wavelet transform filtering systems for noise reduction in ultrasound images

Dominik Vilimek[1], Jan Kubicek[1], Milos Golian[2], Rene Jaros[1], Radana Kahankova[1]*, Pavla Hanzlikova[3], Daniel Barvik[1], Alice Krestanova[1], Marek Penhaker[1], Martin Cerny[1], Ondrej Prokop[4], Marek Buzga[2,5]

**1** Department of Cybernetics and Biomedical Engineering, Faculty of Electrical Engineering and Computer Science, VSB - Technical University of Ostrava, Ostrava, Czech Republic, **2** Human Motion Diagnostic Center, Department of Human Movement Studies, University of Ostrava, Ostrava, Czech Republic, **3** Department of Imaging Method, Faculty of Medicine, University of Ostrava, Ostrava, Czech Republic, **4** MEDIN, a.s., Nove Mesto na Morave, Czech Republic, **5** Deparment of Physiology and Pathophysiology, Faculty of Medicine, University of Ostrava, Ostrava, Czech Republic

* radana.kahankova@vsb.cz

**Data Availability Statement:** All relevant data are within the paper.

**Funding:** M.C. received funding from project entitled Innovative Therapeutic Methods of

## Abstract

Wavelet transform (WT) is a commonly used method for noise suppression and feature extraction from biomedical images. The selection of WT system settings significantly affects the efficiency of denoising procedure. This comparative study analyzed the efficacy of the proposed WT system on real 292 ultrasound images from several areas of interest. The study investigates the performance of the system for different scaling functions of two basic wavelet bases, Daubechies and Symlets, and their efficiency on images artificially corrupted by three kinds of noise. To evaluate our extensive analysis, we used objective metrics, namely structural similarity index (SSIM), correlation coefficient, mean squared error (MSE), peak signal-to-noise ratio (PSNR) and universal image quality index (Q-index). Moreover, this study includes clinical insights on selected filtration outcomes provided by clinical experts. The results show that the efficiency of the filtration strongly depends on the specific wavelet system setting, type of ultrasound data, and the noise present. The findings presented may provide a useful guideline for researchers, software developers, and clinical professionals to obtain high quality images.

## Introduction

Ultrasonography is one of the most used diagnostic imaging methods. This method provides high comfort for the patient since it is non-invasive and thus painless, offers fast real-time, and relatively inexpensive results. Moreover, patients are not exposed to ionizing radiation, making the procedure safer than common medical imaging modalities, such as X-ray [1, 2]. Disadvantages of ultrasonography include the fact that the resulting image quality is operator and patient dependent and also affected by considerable amount of noise. Furthermore, the noise makes the ultrasound examination considerably more complicated, because clinical features

Musculoskeletal System in Accident Surgery within the Operational Programme Research, No. CZ.02.1.01/0.0/0.0/17_049/0008441, R.K. received funding from Ministry of Education of the Czech Republic under Project SP2022/34 and M.P. received funding from Ministry of Education of the Czech Republic under Project SV4502261/SP2022/98 Biomedical Engineering Systems XVIII. The funders had no role in study design, data collection and analysis, decision to publish, or preparation of the manuscript.

**Competing interests:** The authors have declared that no competing interests exist.

are hardly readable and thus this diagnostic tool must be used by highly skilled personnel with specific experience in given field [3, 4]. In addition, the presence of the noise complicates image processing tasks such as object detection, pattern recognition or segmentation. Therefore in recent years, and especially in the last decade, there has been intensifying research effort in the field of image processing with the aim to speed up the examination time and provide more accurate diagnostic information by increasing the quality of the acquired image [5–7].

From the general point of view, the image noise represents a significant phenomenon, which would contribute to image deterioration. It may lead to misinterpretation of clinical outcomes. Technically, the resulting pixels or voxels magnitude is composed of the native clinically important information and noise contribution [8–10]. Hence, these components cannot be completely separated. By applying the image smoothing procedure, we are aimed to at least partially suppress the noise component, and at the same time keep the clinical information. For this reason, we usually search for a compromise between non-distorting clinical information and at the same time elimination as much noise level as possible [11, 12].

There are three noise types typical for ultrasound (US) imaging: Speckle noise, Gaussian noise, and Salt and pepper noise. Speckle noise is the most characteristic and prevalent one, it can affect important image details and may influence the intensity parameters, such as contrast. Gaussian noise is caused by sensor or electronic circuit noise. Salt and Pepper noise occurs due to sudden changes in an image, such as memory cell failure, synchronization error during digitalization or improper function of the sensor cells.

The presence of the above mentioned noise types generally leads to degradation of visual US image quality [13]. Thus, it is important to test the efficacy of the denoising procedure on various types of noise. In this paper, we focus on the image preprocessing, where we often employ so-called image enhancement methods for US image noise-canceling. The preprocessing methods are aimed at noise removal and include mathematical algorithms, which can at least partially reduce the noise from US images. Image preprocessing has a substantial importance for further steps of image processing, including identification and extraction of objects of interest from ultrasound images. Images corrupted with noise or artifacts deteriorate the pixels distribution, thus decreasing performance of the image segmentation techniques such as regional and semantic segmentations [14, 15].

In this case, the segmentation map usually contains blobs, representing the pixel's clusters which do not have an origin in a native image. Such phenomenon is denoted as over-segmentation. Of course, there are other areas, where data smoothing plays an important role, such as the performance of classification techniques or feature extraction [16–18].

Many noise reduction techniques have been developed that preserve the important details in the ultrasound image [19, 20]. The filters working in the spatial domain are applied directly in the spatial image area. A comparative analysis of various ultrasound denoising techniques can be found in [21]. A specific type of spatial filter is the adaptive filtering. Such methods are based on the fact of assigning of weighting coefficients for pixels in a given searching window. Their great advantage is that they do not significantly effect the image edges [22–25]. Among the adaptive filters, the median [26] and bilateral filters are frequently used. Another filter in this category is Rayleigh Maximum Likelihood (RLM) filter [27, 28]. It is important to notice that the performance of these filters are linked to the selection and size of the local window which could significantly differ between datasets. The further category of US denoising filters is based on the principle of diffusion such as speckle Reducing Anisotropic Diffusion filter (SRAD) [29], Modified Anisotropic Diffusion (MSRAD) [30] and similar modifications, like Detail Preserving Anisotropic Diffusion (DPAD) filter [31]. Further, among the denoising techniques are transform domain filters. Such filters firstly transform image and apply despeckling operation in the transformed domain. Here, we recognize thresholding-based

methods [32–34], coefficient correlation-based techniques and Bayesian estimation-based techniques [35–37]. Moreover, currently popular approach is the use of machine and deep learning methods such as Convolutional Neural Networks (CNN) [38–40], Residual Learning Network (ResNet) [41] or Feature-guided Denoising Convolutional Neural Network (FDCNN) [42]. However, these methods are very complex and require a relatively large sample of representative clean data for training [43].

Wavelet-based methods seem to be very effective due to its versatility, relatively simple implementability and good noise reduction capability at higher noise levels [43–46]. However, this technique is still complicated to handle because of plenty various settings through mother's wavelets, levels of decomposition, and other parameters. Therefore, we should be aware of certain limitations regarding using WT in the context of variable settings. Frequently, we must decide a proper wavelet settings not just for a particular ultrasound image data application. Thus, primarily we need to select a suitable procedure, which will effectively perform noise reduction, and at the same time it does not deteriorate the pixels distribution. Another important aspect of each setting is its robustness, i.e. the stability of respective wavelet settings when noise with various intensity level is present. In this context, it is worth to analyze the effect of the wavelet base selection on the filtration efficiency. Such analysis would provide the benefits of evaluating the performance of a suitable wavelet setting for the use in medical imaging.

In this paper, we present a comparative analysis of wavelet performance on real medical ultrasound images. In our study we are capable of batch ultrasound image processing upon the characteristic noise influence with dynamical intensity controlled by noise parameters. Extensive analysis is performed for a specific wavelet settings and each experiment is evaluated using the evaluation metrics such as Mean Squared Error (MSE), correlation coefficient (Corr coeff.), Peak Signal-to-Noise Ratio (PSNR), Structural Similarity Index (SSIM) and Universal Image Quality Index (Q-index) proposed by Wang and Bovik [47]. We selected 8 commonly used wavelets (both Daubechies and Symlets) with different set of decomposition. We provide testing for various ultrasound datasets, including the image data of the musculoskeletal system, abdominal, neck, and carotid. To test the highest amount of noise types and filter settings possible, we corrupted created dataset with noise generators to simulate various image impairments occurring in US images (Speckle, Gaussian, and Salt and Pepper noise). This way, we used the real US images (serving as ground truth) and were able to objectively evaluate the outputs. Our paper is organised as follows: firstly, we describe the used methods, then we show are results of the analysis where we mainly focused on musculoskeletal ultrasound data and finally, we discuss our achievements and future directions.

## Materials and methods

To analyze the effect of the wavelet base selection on the filtration efficiency, we propose an experimental tool for a simultaneous processing of the batch ultrasound images. Herein, we will distinguish types of implemented noise for an image deterioration and used evaluation metrics. Mainly, we used Daubechies and Symlet wavelet families, nevertheless it could be applicable for any wavelet setting and type of image noise. Fig 1 shows a block diagram illustrating the proposed methodology.

### Image acquisition

All experiments were carried out according to ethics approval obtained from the Ethics and Research Committee of University of Ostrava (Ref. No.: $OU - 23913/90 - 2021$). All measurements were made under medical supervision and participants provided written informed consent form prior to data collection. All the information is processed anonymously. All data were

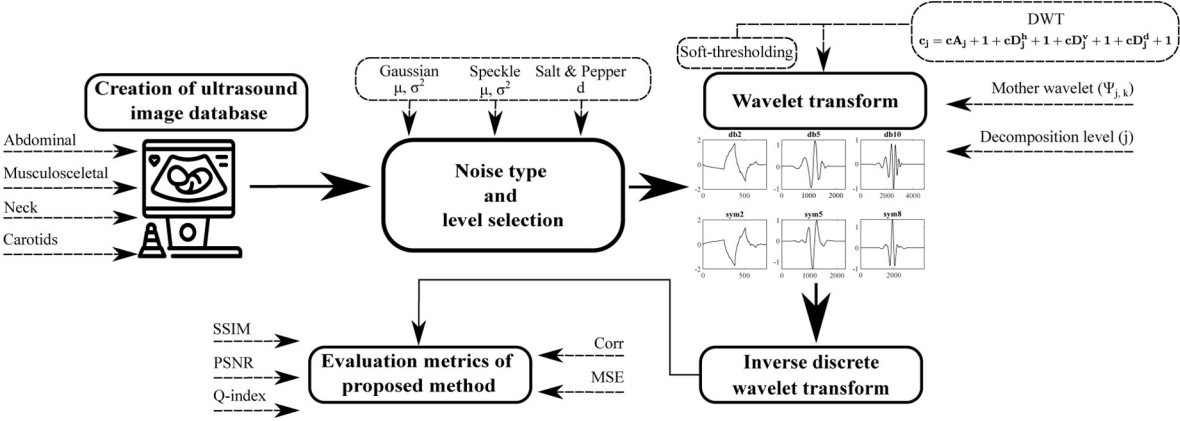

**Fig 1. A simplified diagram of an experimental environment for testing and evaluation of dynamical wavelet behavior.**

acquired by an ultrasonographic device GE LOGIQ P6 PRO using standard ultrasound probes. For the purpose of this study, we created a database containing in total of 292 ultrasound images. The database is divided into four categories based on the examined area: *Abdominal*—100 images, *neck*—40 images, *carotid*—80 images and *musculoskeltal* (MSK)—72 images, contain images of knee, tendons and shoulders. All images were recorded in B-mode with resolution of 512x512 pixels. The database contains both physiological and pathological images, see example in Fig 2.

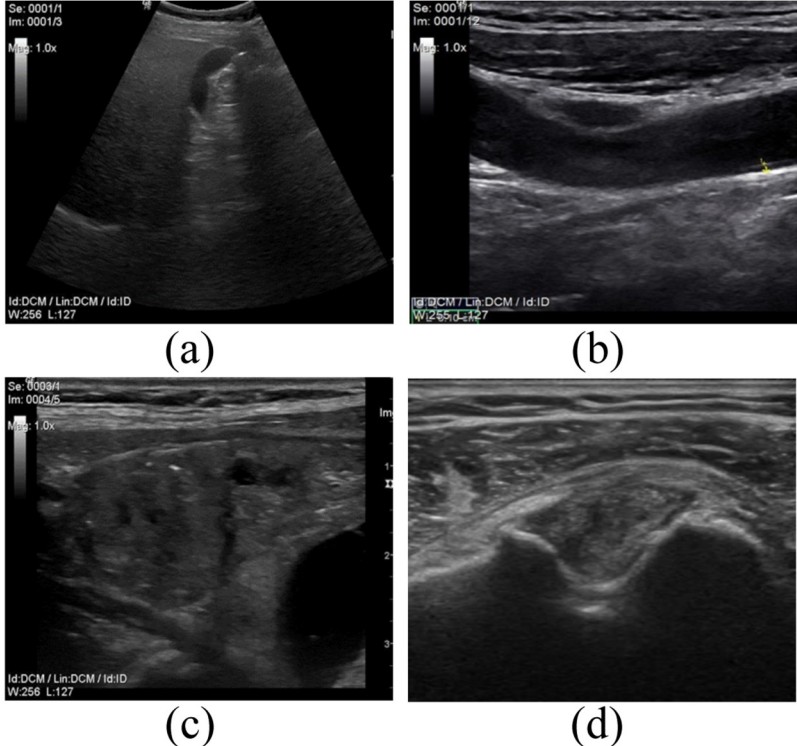

**Fig 2. Examples of the ultrasound images from all four cathegories of the created database: Abdominal (a), Carotids (b), Neck (c), and Musculoskeletal system (d).**

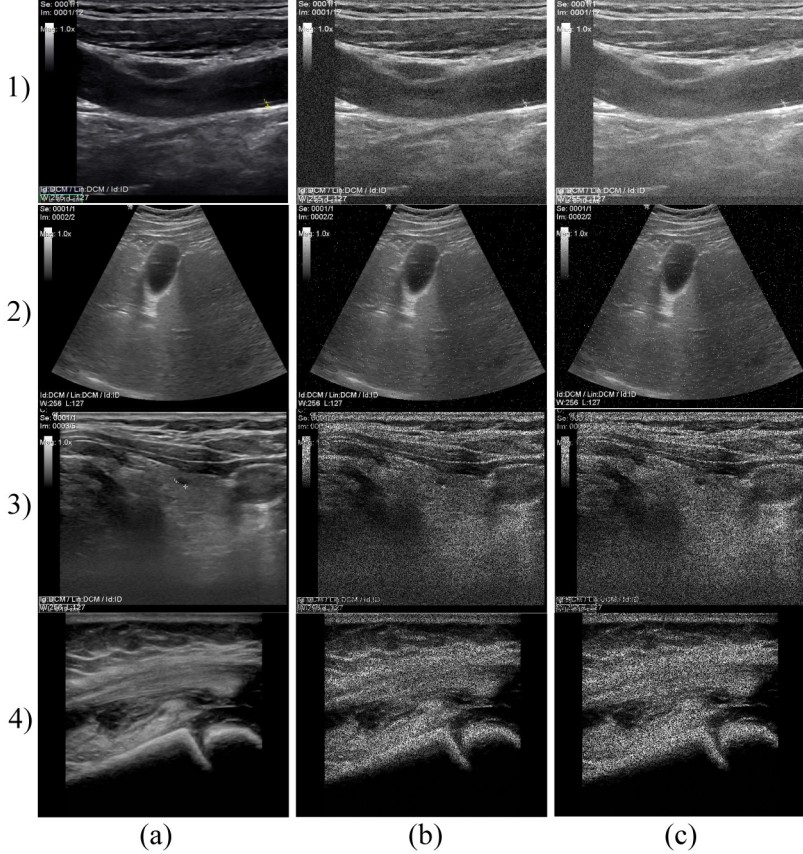

(a) (b) (c)

**Fig 3.** Examples of native images from different databases (a) and deteriorated by various types and levels of noise (b, c). 1) Gaussian noise applied on carotid ultrasound images: (b) Gaussian noise level ($\sigma^2 = 0.01$, $\mu = 0.1$) and (c) Gaussian noise level ($\sigma^2 = 0.01$, $\mu = 0.2$); 2) Salt and Pepper noise applied on abdominal ultrasound images: b) Salt and Pepper noise density ($d = 0.1$) and c) Salt and Pepper noise density ($d = 0.2$); 3) and 4) speckle noise applied on neck (3) and MSK (4) ultrasound images: b) variance of speckle noise ($\sigma^2 = 1$) and c) variance of speckle noise ($\sigma^2 = 2$).

## Artificial image deterioration

To perform an analysis of the wavelet base selection on the filtration efficiency, we applied various types and level of noise to simulate the effect of image deterioration. Such synthetic noise produce a specific image with variable noise intensity, set by related noise parameters. We used additive Gaussian, impulse salt and pepper and multiplicative speckle noise. We applied 20 different levels with various parameters that are dependent on the type of noise, mainly set by mean, variance, and noise density.

Gaussian noise was set on a default variance ($\sigma^2 = 0.01$) and mean values $\mu = \{0.01, 0.02, \ldots, 0.20\}$. An example of ultrasound image degraded by Gaussian noise is presented in Fig 3 1). Salt and Pepper noise, determined by the noise density $(d)$ was set on $d = \{0.001, 0.002, \ldots, 0.02\}$. Example is shown in Fig 3 2). The design of speckle noise determined by the mean value and variance was set on a constant value $\mu = 0$ and 20 various levels $\sigma^2 = \{0.1, 0.2, \ldots, 2\}$, see example in Fig 3 3) and 4).

## Design of denoising system

Firstly, we had to choose a suitable type of mother wavelet function that is in general, chosen empirically based on the characteristics of the signal. Mother wavelet is a prototype for

generating the other window functions. In our analysis, we present a comparison of various kinds of Daubechies (*db2*, *db5*, *db10*, *db15*, *db20*, *db22*, *db25*, and *db30*) and Symlets (*Sym2*, *Sym3*, *Sym5*, *Sym10*, *Sym15*, *Sym20*, *Sym25*, and *Sym29*). Then, the ultrasound data are gradually decomposed into approximation and detail coefficients based on the level of decomposition. Previous evidence shows that higher level of decomposition does not bring to a significant improvement to denoising performance [46, 48]. However, the selection of the mother wavelet can be affected by setting the level of decomposition. Moreover, the wavelet-based methods with lower level of decomposition, is not sufficient to reduce the noise with the higher level. Therefore, a higher level of decomposition is desirable, so it was set to 3 and 4. Then, the soft-thresholding method is used by the Birgé-Massart strategy [49] for its ability to preserve the image edges during image filtration. To analyze wavelet-based performance, each mother's wavelet was objectively compared with the gold standard (i.e. native images, where we suppose the presence of a neglectable level of image noise). Through this procedure, we were able to provide a large quantitative analysis of wavelet filtration efficiency based on the selection of mother's wavelets.

## Objective evaluation of wavelet's setting

The medical image quality could be evaluated by means of different methods based on the specific criteria, such as the diagnostic quality of the image or its other characteristics (contrast, blur, noise or sharpness). For this analysis, we used various evaluation metrics, such as MSE, PSNR, Q-index, SSIM, and correlation index. These parameters are defines as follows:

- **Correlation coefficient** returns a correlation between arrays A and B, that is in the interval 0 to 1, where 1 represents a complete correlation.

- **MSE** is the average squared difference between two data samples is measured, e.g., the reference and degraded image. The smaller the mean squared error, the closer is best fit. If the image is defined in the *MxN* domain, MSE is defined as follows:

$$MSE = \frac{1}{M \cdot N} \cdot \sum_{i=1}^{M} \sum_{j=i}^{N} (g_{i,j} - f_{i,j})^2, \tag{1}$$

where $g_{i,j}$ denotes the original ultrasound image and $f_{i,j}$ denotes the noisy image. MSE is widely used to compare image quality, however if its used alone it does not provide a sufficient correlation of reasonable quality, therefore should be used with other metrics or visual assessment.

- **PSNR** represents the ratio of the maximum possible signal power to the distortion power, which will be higher for a better image and vice versa. It measures how exactly the transformed image resembles the original. PSNR can be determined as follows:

$$PSNR = 10 \log \frac{D^2}{MSE}, \tag{2}$$

where $D$ represents the dynamical intensity range (e.g. fof 8-bit US image, it is 256 gray levels).

- **Q-index** measures any image distortion as a combination of a loss of correlation, intensity and contrast distortion. These factors, defining Q-index can be interpreted by the following

way:

$$Q = \frac{\sigma_{xy}}{\sigma_x \sigma_y} \cdot \frac{2\bar{x}\bar{y}}{\bar{x}^2 + \bar{y}^2} \cdot \frac{2\sigma_x \sigma_y}{\sigma_x^2 + \sigma_y^2}, \tag{3}$$

where the first component of the equation denotes the correlation coefficient between $x$ and $y$, which describes the native image. This coefficient measures the degree of linear correlation between these coefficients and their dynamic range $[-1, 1]$. The second component of the equation in the range $[0, 1]$ measures the link between mean brightness of $x$ and $y$.

- **SSIM** performs modeling of the structural information of an image which is based on fact that the pixels of the natural image show strong dependencies providing useful information about its structure. The SSIM algorithm defines image degradation as a structural change and performs measurements of similarity in three steps, comparison by intensity, contrast and image structure. The SSIM is in the range $[0, 1]$, where value 1 could be achieved only if image $x$ is identical to image $y$. SSIM index can be defined by the following equation:

$$SSIM = \frac{(2\bar{x}\bar{y} + c_1) \cdot (2\sigma_{xy} + c_2)}{(\bar{x}^2 + \bar{y}^2 + c_1) \cdot (\sigma_x^2 + \sigma_y^2 + c_2)}, \tag{4}$$

where

$$\bar{x} = \frac{1}{L}\sum_{i=1}^{L} x_i, \quad \bar{y} = \frac{1}{L}\sum_{i=1}^{L} y_i,$$

$$\sigma_x^2 = \frac{1}{L-1}\sum_{i=1}^{L}(x_i - \bar{x})^2, \quad \sigma_y^2 = \frac{1}{L-1}\sum_{i=1}^{L}(y_i - \bar{y})^2,$$

$$\sigma_{xy} = \frac{1}{L-1}\sum_{i=1}^{L}(x_i - \bar{x}) \cdot (y_i - \bar{y}), \tag{5}$$

in those equations $x$, $y$, and $L$ represent the original image, the test image, and the number of pixels in the image, respectively. Moreover, $c_1$ and $c_2$ are the defined values used to calculate the SSIM metric for stabilization. Parameters $\sigma_x$, $\sigma_y$ represent variance of the signal sample $x$ and $y$, where $\sigma_{xy}$ corresponds to the mutual connection between $x$ and $y$.

## Results and analysis

The results of wavelet-based noise suppression system for a comparative analysis of various mother wavelet selection are based on objective metrics (correlation coefficient, MSE, PSNR, Q-index and SSIM). Moreover, the computational demands of the proposed noise-canceling algorithm was measured. As important clinical aspects must be taken into account when applying noise-canceling procedure, the radiologist view is provided. The graphs below show results of analysis from the musculoskeletal database. The series of analyses investigated the filtration efficiency of Dabechies and Symlets wavelets family on real US images degraded by Gaussian, salt and pepper and speckle noise.

## Elimination of Gaussian noise

When assessing the quality of the filtration using different objective metrics, various observations can be made. In all experiments, the highest efficacy was achieved at lowest level of Gaussian noise ($\sigma^2 = 0.01$, $\mu = 0.01$); the higher the noise level, the lower efficacy.

In terms of correlation coefficient, we can notice linear dependence between increasing noise levels and filtration effectiveness in level 3 and 4 for both Daubechies and Symlet families (see Figs 4a, 4b, 5a and 5b, respectively). The most effective wavelets for both decomposition levels were *Db5* and *Sym15*, while the latter slightly outperforming the other tested wavelets.

When assessing the filtration efficacy using MSE, one can notice exponential dependence between noise level and the quality of filtration. All of the tested wavelets from both families achieved similar results for level 3 and 4 of decomposition (see Figs 4c, 4d, 5c and 5d). The differences between the individual system settings are nearly indistinguishable, as illustrated by the zoomed sections in corresponding figures.

As for PSNR parameter, the dependence between the filtration quality and noise level is nearly linear. The differences between individual wavelets are noticeable only at lower noise levels, see zoomed sections in Figs 4e, 4f, 5e and 5f. The best results were achieved by *Db5* and *Sym3* at decomposition level 3.

The evaluations using the Q-index parameter show nearly linear dependence between noise levels and filtration efficacy. Again, the differences between individual system settings vary only for low noise levels, see zoomed sections in Figs 4g, 4h, 5g and 5h.

In case of SSIM evaluation parameter, the results show that the filtration efficacy is linearly dependent on the image's noise level. The higher the noise the lower the efficacy. For both tested decomposition levels (3 and 4) in Daubechies wavelet family, the *Db10* and *Db25* wavelet showed the best results while *Db2* showed the worst results. As for Symlet wavelet family, the results shows higher effectiveness within various image impairments. The best results for both tested decomposition levels (3 and 4) were achieved for *Sym15* and *Sym20* wavelets. Contrary, the lowest efficiency was achieved using the wavelet *Sym2*.

Fig 6 1) and 2) show the efficiency of the denoising system using the *Db5* with decomposition level 3 and *Db5* with decomposition level 4, respectively. With increasing level of decomposition, we can observe a loss of significant anatomical details from USG images as notable in Fig 6. Similarly, by using the *Sym15* and *Sym2*, see Fig 6 3) and 4), respectively.

Table 1 show median comparison of SSIM and correlation coefficient to determine the efficiency using the *Db5* wavelet with level od decomposition 3 for all noise levels on different datasets. Interestingly, we can notice slight variability between individual datasets, see SSIM for abdominal dataset 0.401 and MSK 0.566.

## Elimination of Salt and Pepper noise

The results below show a similar trend as for Gaussian noise, i.e. in all experiments the higher the noise level, the lower efficacy. However, contrary to results obtained with the Gaussian noise, the efficacy achieved varies with higher noise levels. Also, the dependency between the noise levels and the efficacy of the filtration varies among the different wavelet types, as for some it is nearly linear (e.g. Db2 in Fig 7 or Sym2 in Fig 8) and logarithmic for the others.

Although we can see an overall decrease in efficiency of all tested wavelets with the higher decomposition level, the *Db30*, *Db25*, *Db22*, and *Db20* appear to be more effective in comparison with the rest of the tested wavelet types, and could thus enable a better preservation of relevant diagnostics information, lower loss of contrast, and edge preservation. The lower efficacy decrease with higher noise is also a sign of the system robustness. Contrary to the tests on

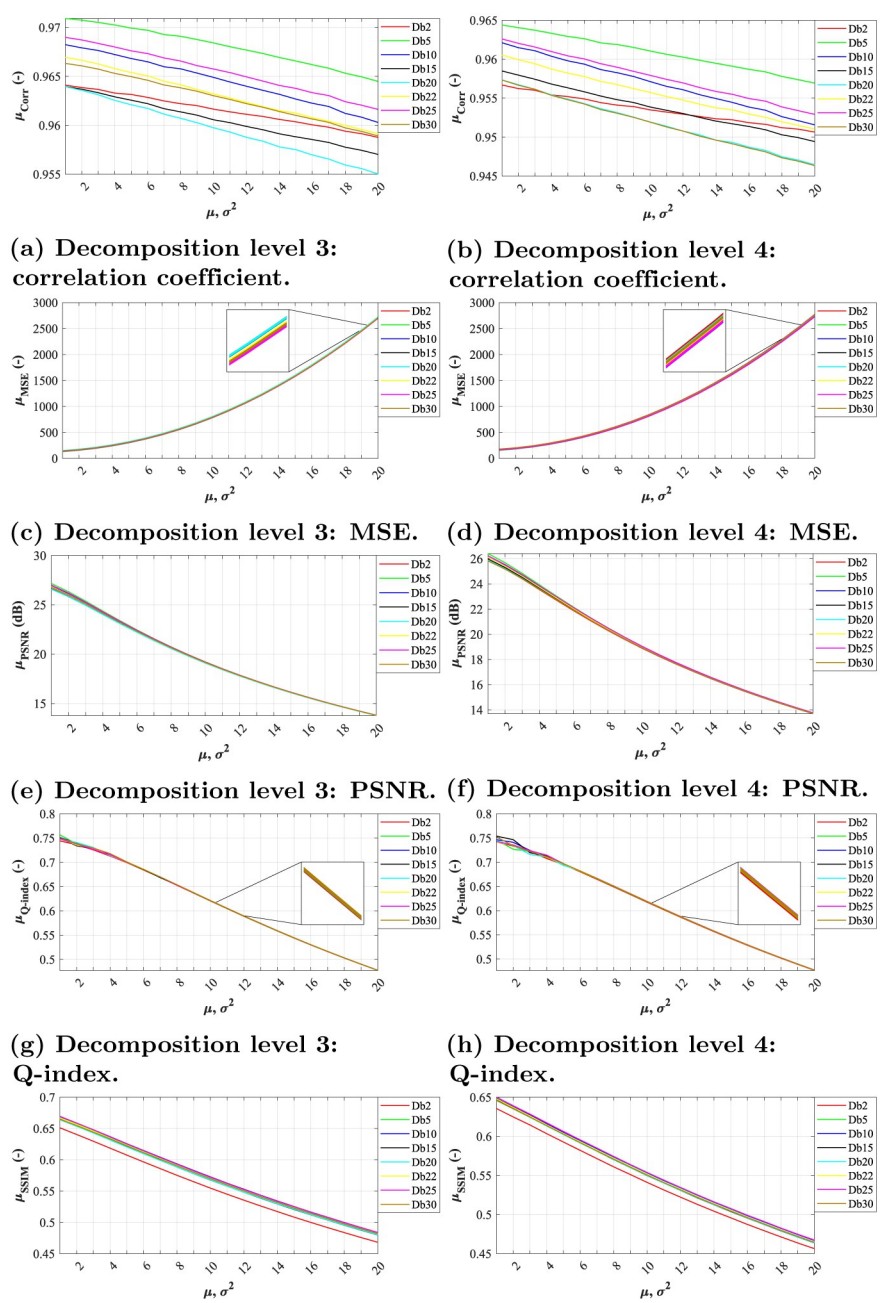

**Fig 4. A comparative analysis of Gaussian noise ($\sigma^2 = 0.01$, $\mu = \{0.01, 0.02, \ldots, 0.20\}$) for Daubechies wavelets with decomposition level 3 and 4.** (a) Decomposition level 3: correlation coefficient. (b) Decomposition level 4: correlation coefficient. (c) Decomposition level 3: MSE. (d) Decomposition level 4: MSE. (e) Decomposition level 3: PSNR. (f) Decomposition level 4: PSNR. (g) Decomposition level 3: Q-index. (h) Decomposition level 4: Q-index. (i) Decomposition level 3: SSIM. (j) Decomposition level 4: SSIM.

Gaussian noise, the *Db5* appear to be the least effective wavelet used along with the *Db2*, see Fig 7.

The analysis of the Symlet family shows a similar efficiency as in the analysis performed on images degraded by Gaussian noise, see Fig 5. The efficiency assessed using the objective

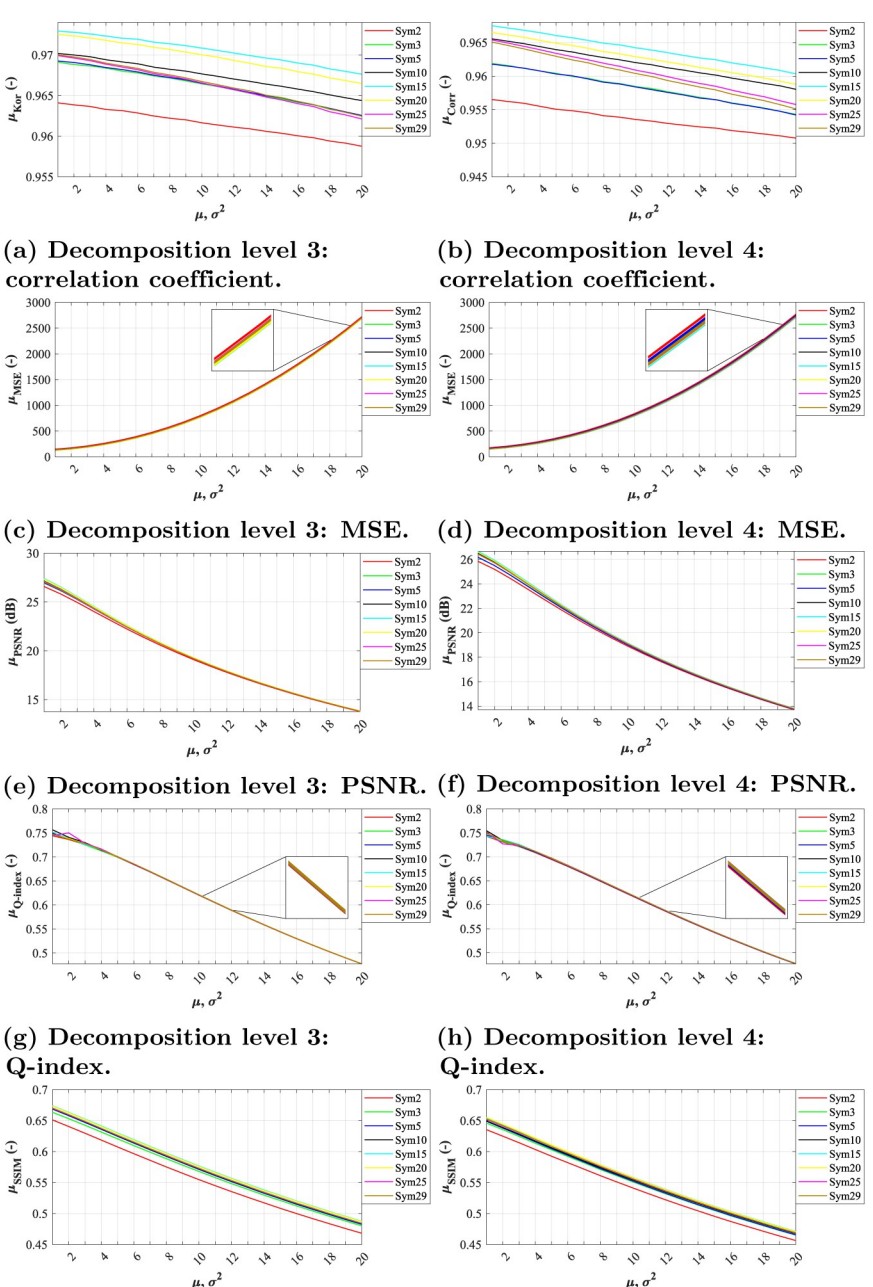

(a) Decomposition level 3: correlation coefficient.

(b) Decomposition level 4: correlation coefficient.

(c) Decomposition level 3: MSE.

(d) Decomposition level 4: MSE.

(e) Decomposition level 3: PSNR.

(f) Decomposition level 4: PSNR.

(g) Decomposition level 3: Q-index.

(h) Decomposition level 4: Q-index.

(i) Decomposition level 3: SSIM.

(j) Decomposition level 4: SSIM.

**Fig 5. A comparative analysis of Gaussian noise ($\sigma^2 = 0.01$, $\mu = \{0.01, 0.02, \ldots, 0.20\}$) for Symlet wavelets with decomposition level 3 and 4.** (a) Decomposition level 3: correlation coefficient. (b) Decomposition level 4: correlation coefficient. (c) Decomposition level 3: MSE. (d) Decomposition level 4: MSE. (e) Decomposition level 3: PSNR. (f) Decomposition level 4: PSNR. (g) Decomposition level 3: Q-index. (h) Decomposition level 4: Q-index. (i) Decomposition level 3: SSIM. (j) Decomposition level 4: SSIM.

parameters of the *Sym29* and *Sym15* is similar. Contrary *Sym2* and *Sym3* appear to be significantly less efficient, as demonstrated by the results depicted in Fig 8. Further analysis shows that a higher level of decomposition leads to obtaining a blurry image. However, the results demonstrate that the Symlets are less effective at denoising of Salt and Pepper noise. In Fig 9

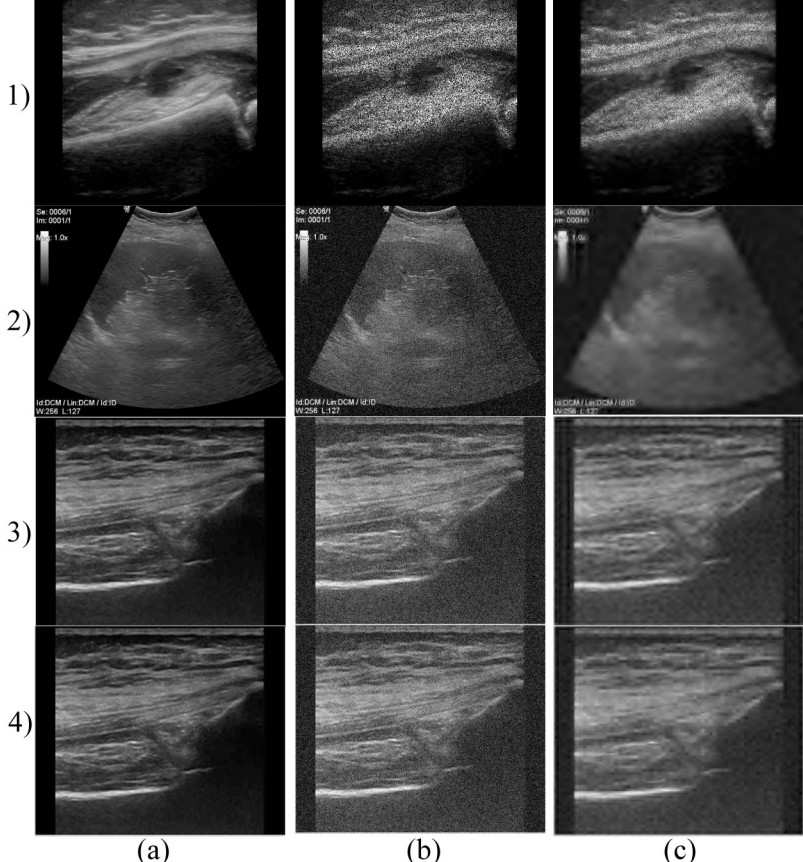

**Fig 6. An example of denoising results on different types of datasets using the selected wavelets.** 1) Denoising of carotid images using the *Db5*, level of decomposition 3. a) native image, b) noisy image (Gaussian noise ($\sigma^2 = 0.01$, $\mu = 0.05$.)) and c) result of noise-canceling procedure; 2) Denoising of abdominal images using *Db5*, level of decomposition 4. a) native image, b) noisy image (Gaussian noise ($\sigma^2 = 0.01$, $\mu = 0.05$.)) and c) filtration result; 3) and 4) Denoising of MSK images using the *Sym15* (3) and *Sym2* (4), level of decomposition 3. a) native image, b) noisy image (Gaussian noise ($\sigma^2 = 0.01$, $\mu = 0.05$.)) and c) filtered MSK image.

(3) (4) we can see that the noise was not completely suppressed as with Daubechies family, see Fig 9 (1) (2). Table 2 show a comparison of the *Db30* efficiency using the median for all implemented noise levels and datasets.

In terms of correlation coefficient, we can notice linear dependence between increasing noise levels and filtration effectiveness in level 3 and 4 for both Daubechies and Symlet families

**Table 1. Median comparison of SSIM and correlation coefficient for *Db5* wavelet with decomposition level 3 and various noise levels and datasets.**

|  | Corr. coeff. | | | SSIM | | |
|---|---|---|---|---|---|---|
|  | **Gaussian** | **S&P** | **Speckle** | **Gaussian** | **S&P** | **Speckle** |
| Abdominal | 0.932 | 0.919 | 0.815 | 0.401 | 0.726 | 0.581 |
| Carotids | 0.931 | 0.920 | 0.812 | 0.432 | 0.759 | 0.689 |
| Neck | 0.915 | 0.930 | 0.846 | 0.567 | 0.749 | 0.688 |
| MSK | 0.968 | 0.934 | 0.866 | 0.566 | 0.751 | 0.667 |
| Mean | 0.937 | 0.926 | 0.835 | 0.492 | 0.746 | 0.656 |

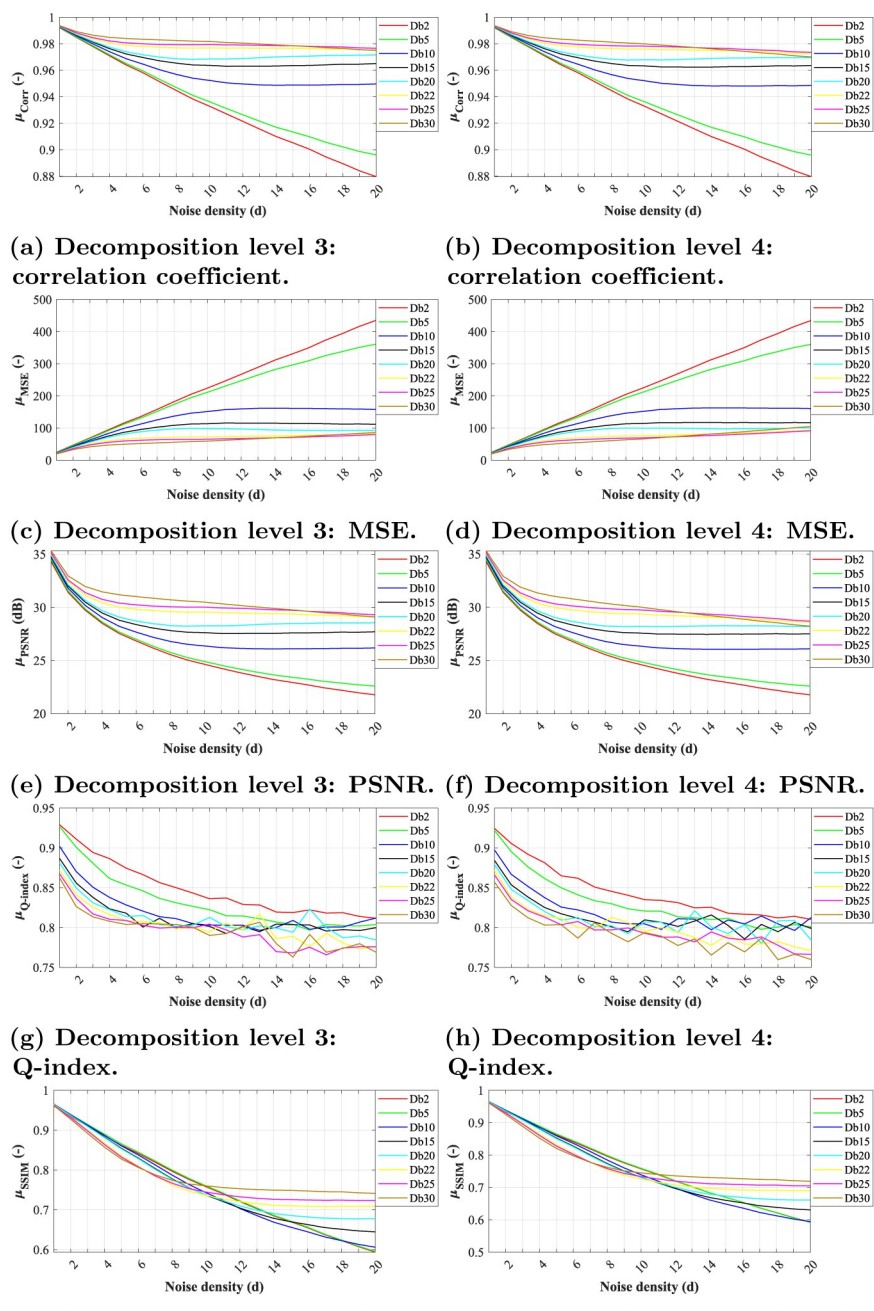

**Fig 7. A comparative analysis of Salt and Pepper noise ($d$ = {0.001, 0.002, ..., 0.02}) for Daubechies wavelets with decomposition level 3 and 4.** (a) Decomposition level 3: correlation coefficient. (b) Decomposition level 4: correlation coefficient. (c) Decomposition level 3: MSE. (d) Decomposition level 4: MSE. (e) Decomposition level 3: PSNR. (f) Decomposition level 4: PSNR. (g) Decomposition level 3: Q-index. (h) Decomposition level 4: Q-index. (i) Decomposition level 3: SSIM. (j) Decomposition level 4: SSIM.

(see Figs 7a, 7b, 8a and 8b, respectively). The least effective wavelets for both decomposition levels were *Db2* and *Db5*, while the latter being slightly more effective. The other tested wavelets outperformed them significantly. As for the Symlet family, the most effective wavelets were *Sym25*, *Sym15*, and *Sym29*, especially for growing noise levels.

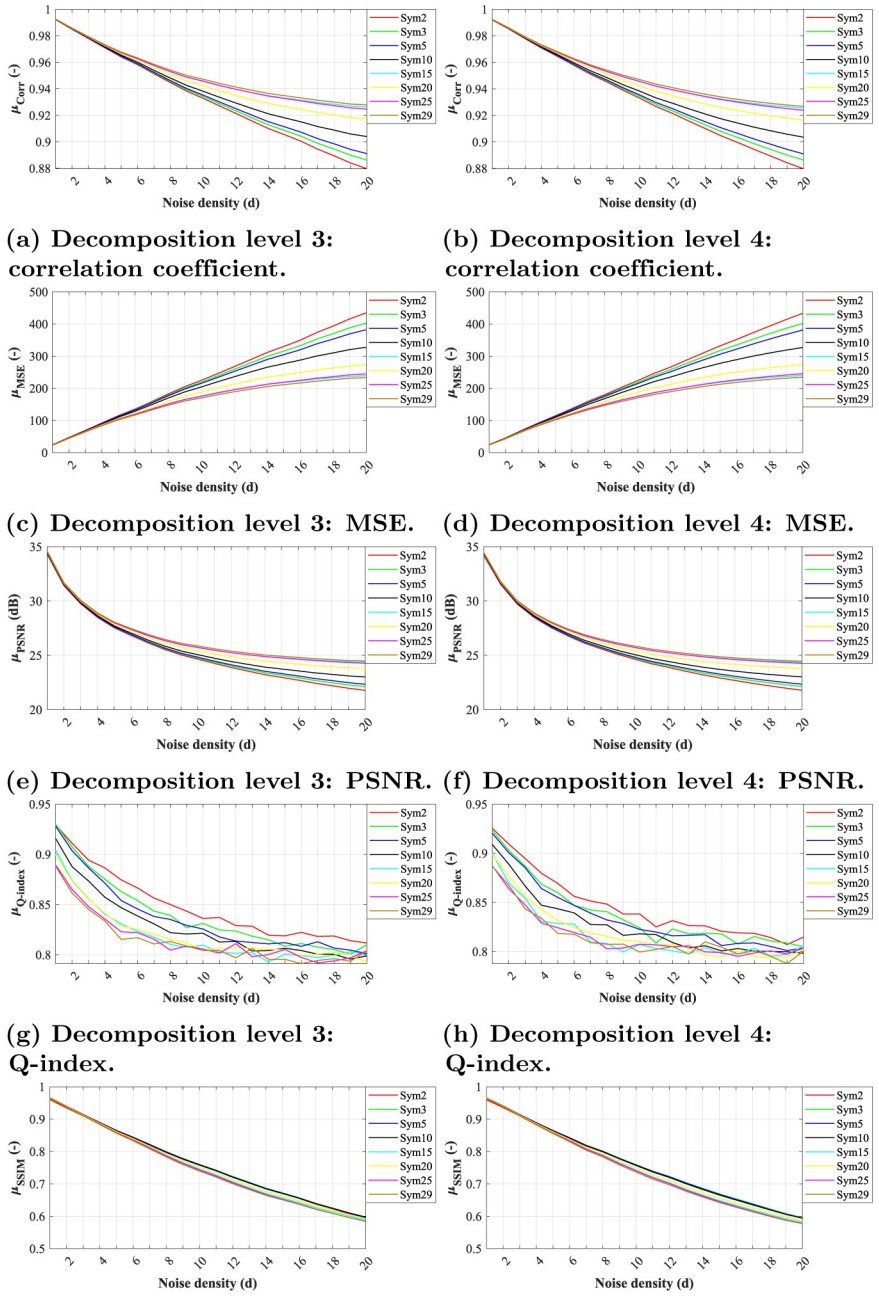

**Fig 8. A comparative analysis of Salt and Pepper noise ($d$ = {0.001, 0.002, . . ., 0.02}) for Symlets wavelets with decomposition level 3 and 4.** (a) Decomposition level 3: correlation coefficient. (b) Decomposition level 4: correlation coefficient. (c) Decomposition level 3: MSE. (d) Decomposition level 4: MSE. (e) Decomposition level 3: PSNR. (f) Decomposition level 4: PSNR. (g) Decomposition level 3: Q-index. (h) Decomposition level 4: Q-index. (i) Decomposition level 3: SSIM. (j) Decomposition level 4: SSIM.

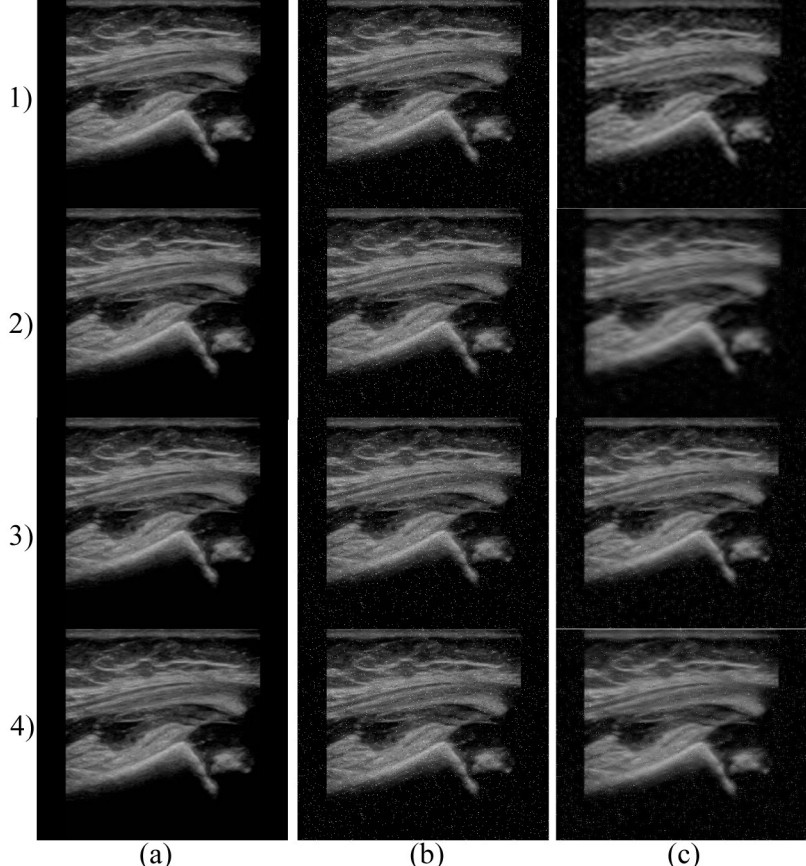

**Fig 9. An example of denoising results on MSK image dataset using the selected wavelets.** 1) Denoising using the *Db25*, level of decomposition 3. a) native image, b) noisy image (Salt and Pepper noise ($d$ = 0.02)) and c) result of noise-canceling procedure; Denoising using the *Db25*, level of decomposition 4. a) native image, b) noisy image (Salt and Pepper noise ($d$ = 0.02)) and c) filtration result; Denoising using the *Sym29*, level of decomposition 3 (3) and *Sym29* with level of decomposition 4 (4). a) native image, b) noisy image (Salt and Pepper noise ($d$ = 0.02)) and c) filtered MSK image.

In Q-index evaluation, one can notice significant fluctuations of the values for higher noise levels for both tested wavelet families (see Figs 7g, 7h, 8g and 8h). This is associated with insta-bility of the filtration system which also has the affect on the resulting image, where we can notice that the filtration is not effective enough—especially in the case of the Symlet wavelets (see Fig 9).

**Table 2. Median comparison of SSIM and correlation coefficient for *Db30* wavelet with decomposition level 3 and various noise levels and datasets.**

| | Corr. coeff. | | | SSIM | | |
|---|---|---|---|---|---|---|
| | **Gaussian** | **S&P** | **Speckle** | **Gaussian** | **S&P** | **Speckle** |
| Abdominal | 0.927 | 0.968 | 0.808 | 0.406 | 0.685 | 0.542 |
| Carotids | 0.925 | 0.967 | 0.827 | 0.430 | 0.731 | 0.672 |
| Neck | 0.911 | 0.964 | 0.846 | 0.567 | 0.754 | 0.678 |
| MSK | 0.968 | 0.981 | 0.857 | 0.565 | 0.758 | 0.637 |
| Mean | 0.931 | 0.970 | 0.834 | 0.490 | 0.732 | 0.632 |

As for the SSIM parameter-based evaluation, there are notable differences between the efficacy of the tested wavelet families. While for Symlets (see fig XY i,j) the dependency between the noise levels and the filtration efficacy is nearly linear for all tested wavelet bases, the Daubechies, the trend is quite different. The dependency is rather logarithmic and some of the wavelet bases (e.g. Db25 and Db30) are less effective for the lower noise density ($d$ = 2–8) than for the rest of the tested wavelet bases while more effective for higher noise density (Db10) it is more effective.

Table 2 show a comparison of the *Db30* efficiency using the median for all implemented noise levels and datasets.

## Elimination of Speckle noise

Finally, analyzes performed on images degraded with speckle noise show that the *Db2* wavelet appears to be the most effective according to all metrics used. Contrary, the *Db25* and *Db30* wavelets appear to reach the worst results, see Fig 10.

Interestingly, Symlets of the same order reach the same results as the Daubechies family. We can notice that the best results are obtained with a *Sym2*, while the lowest efficiency can be attributed to *Sym25* and *Sym29*, see Fig 11. Similarly, with the higher level of decomposition, we can see a more intensive blurring and therefore the resulting images becomes harder to read, see Fig 12. Table 3 shows median values for the *Db2* wavelet.

## Computational complexity of algorithm

The complexity of an algorithm is the amount of resources required to run it. The time that the CPU needs to run of the proposed noise-canceling method was tested on all used databases (abdominal—100, carotids—80, neck—40, musculoskeletal—72 images). The analysis were carried out on a PC with the configuration: quad-core Intel Core i7-7700HQ processor (2.80 GHz, TB 3.8 GHz, HyperThreading); 32 GB RAM DDR4; NVIDIA GeForce GTX 1050 TI. All results are in seconds. Based on Tables 4 and 5, we found that the use of Dabechies was almost 5 times more efficient in terms of computational complexity. We can also notice a slightly increasing time at a higher level of decomposition. However, computing time could change significantly with a better computing unit with GPU processing capability or a more elegant solution such as parallel techniques [50].

## Analysis performed by radiologists

From the clinical point of view, knowledge and interpretation of typical physiological ultrasonographic images of body areas, organs, variants, and pathological changes within various diagnoses is necessary for the interpretation and evaluation of US images. Pathological changes can diffusely affect the entire organ or system of organs, or locally cause a change in the structure and size of parts of the affected organ.

Fig 3 1) is B-mode of the carotid artery where we can evaluate its course and lumen width. In the native representation, the lumen content is anechogenic, the wall is of fine higher echogenicity, and it consists of two fine linear structures. We can evaluate possible pathological changes of the wall that have different echogenicity depending on the content of calcifications.

Fig 3 2) shows an example of abdominal examination in the area of the epigastrium, the dominant image is the parenchyma of the liver, which has a medium echogenic, uniform medium-grained structure with fine diffusely scattered echoes. Hepatic veins can be distinguished from portal veins by increased echogenicity of the periportal ligament. The biliary outlet system is normally lean with an anechogenic content. The gallbladder (with an empty

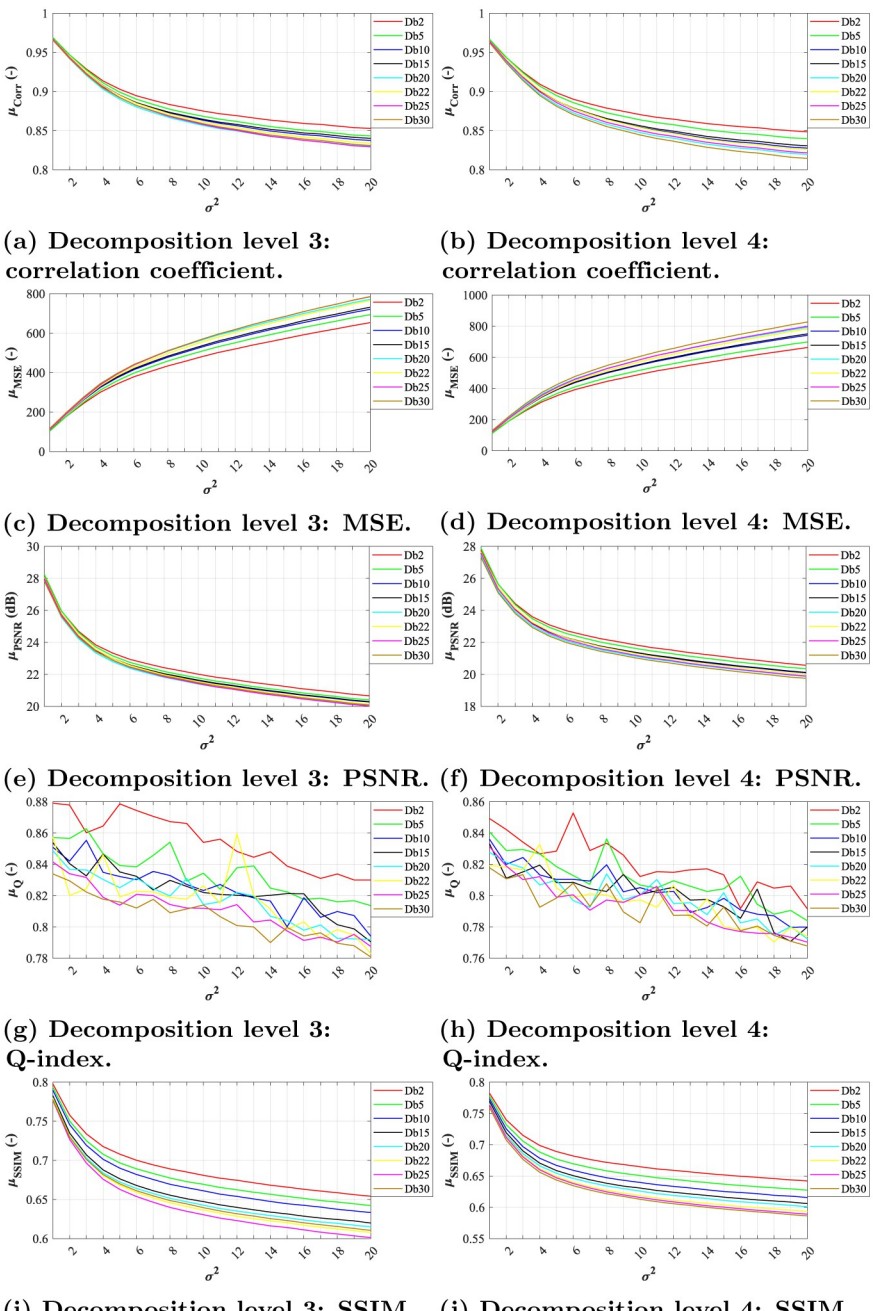

**Fig 10. A comparative analysis of speckle noise ($\mu = 0$, $\sigma^2 = \{0.1, 0.2, \ldots, 2\}$) for Daubechies wavelets with decomposition level 3.** (a) Decomposition level 3: correlation coefficient. (b) Decomposition level 4: correlation coefficient. (c) Decomposition level 3: MSE. (d) Decomposition level 4: MSE. (e) Decomposition level 3: PSNR. (f) Decomposition level 4: PSNR. (g) Decomposition level 3: Q-index. (h) Decomposition level 4: Q-index. (i) Decomposition level 3: SSIM. (j) Decomposition level 4: SSIM.

stomach) preprandially has a cystic character, a homogeneous anechogenic content with a fine single-layer wall. Ultrasonographically, we mainly evaluate the presence of intraluminal pathological content, which is manifested by a change in echogenicity and acoustic tents. Another area of interest for the epigastrium is the parenchymatous organs of the pancreas and spleen,

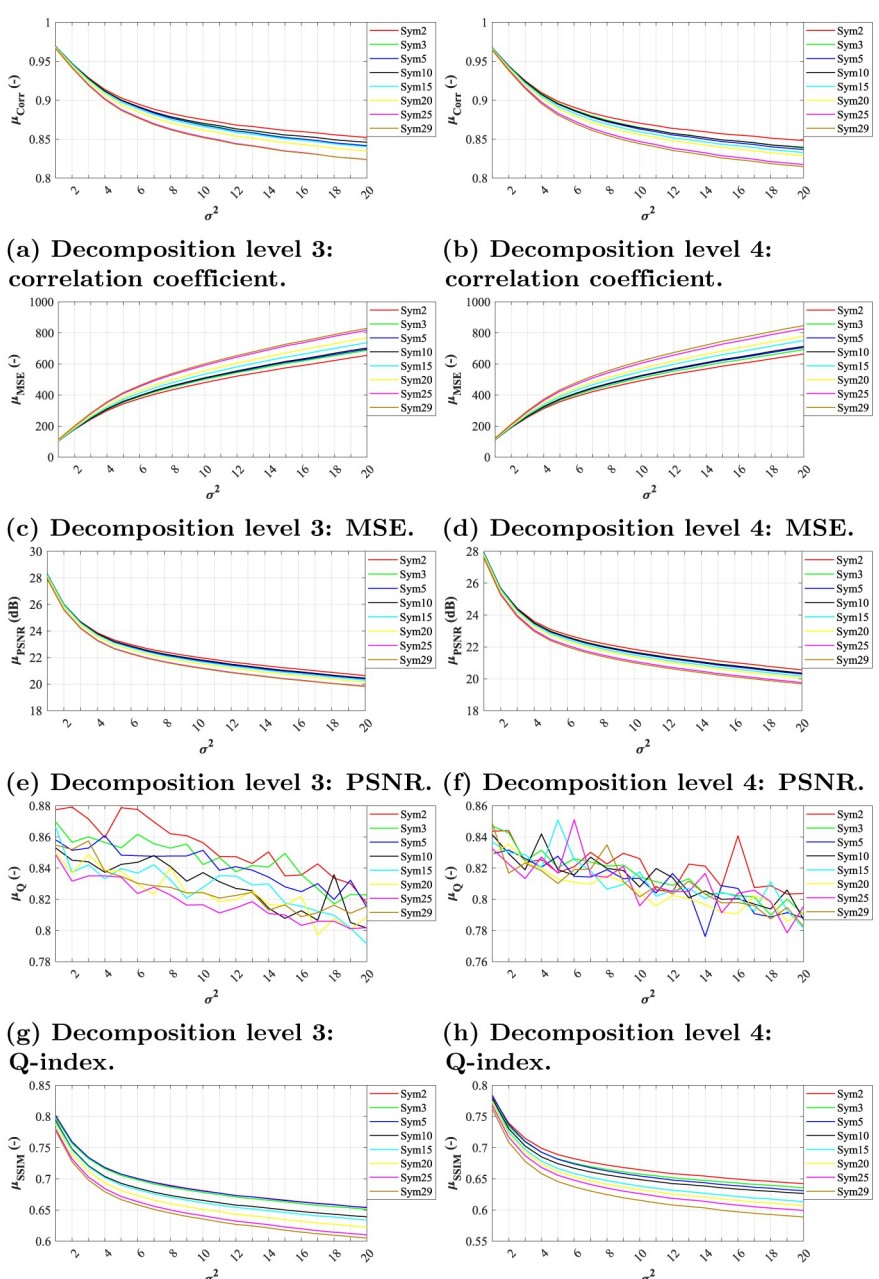

**Fig 11. A comparative analysis of speckle noise ($\mu = 0$, $\sigma^2 = \{0.1, 0.2, \ldots, 2\}$) for Symlets wavelets with decomposition level 4.** (a) Decomposition level 3: correlation coefficient. (b) Decomposition level 4: correlation coefficient. (c) Decomposition level 3: MSE. (d) Decomposition level 4: MSE. (e) Decomposition level 3: PSNR. (f) Decomposition level 4: PSNR. (g) Decomposition level 3: Q-index. (h) Decomposition level 4: Q-index. (i) Decomposition level 3: SSIM. (j) Decomposition level 4: SSIM.

which normally have slightly higher echogenicity and a more homogeneous structure than the parenchyma of the liver.

Fig 3 3) shows a thyroid gland with higher echogenicity, diffusely fine to medium coarse echoes. The lobes of the thyroid gland are located on the sides of the trachea, which form an

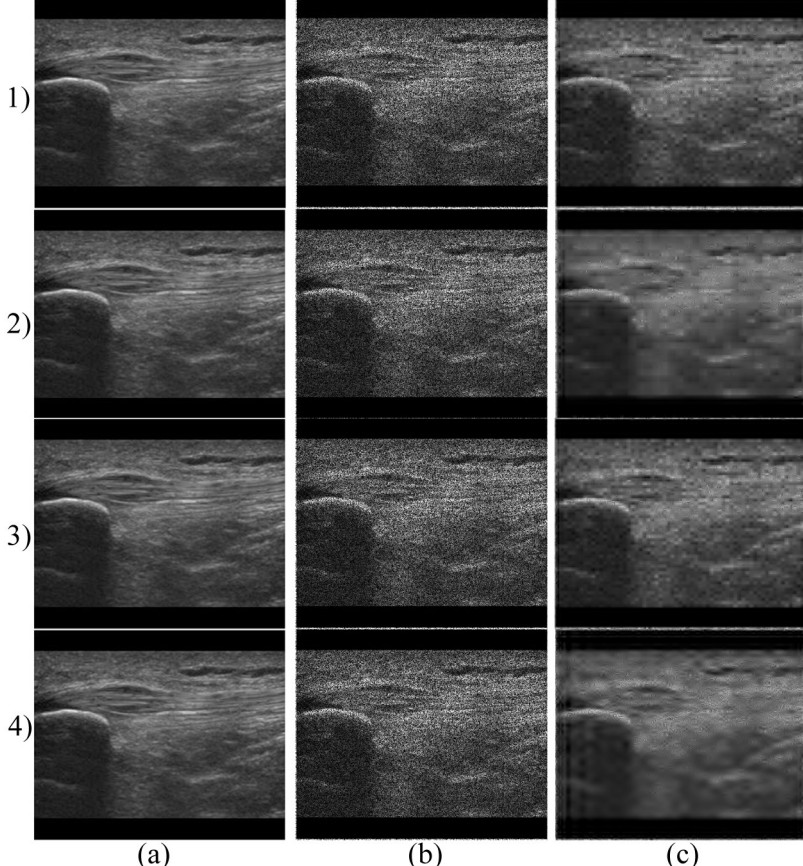

**Fig 12. An example of denoising results on MSK image dataset using the selected wavelets.** 1) Denoising using the *Db2*, level of decomposition 3. a) native image, b) noisy image (speckle noise ($\mu = 0$, $\sigma^2 = 1$)) and c) result of noise-canceling procedure; Denoising using the *Db2*, level of decomposition 4. a) native image, b) noisy image (speckle noise ($\mu = 0$, $\sigma^2 = 1$)) and c) filtration result; Denoising using the *Sym2*, level of decomposition 3 (3) and *Sym29* with level of decomposition 4 (4). a) native image, b) noisy image (speckle noise ($\mu = 0$, $\sigma^2 = 1$)) and c) filtered MSK image.

acoustic shadow of the air column. The surrounding neck muscle and adipose tissue have a linear echostructure, slightly lower echogenicity. Large cervical vessels located laterally have an anechogenic lumen and a fine echogenic wall under normal circumstances. We also display lymph nodes here and evaluate their size, shape, and vascularization. Under normal circumstances, they have an ovoid elongated shape and a slightly lower echogenicity compared to the thyroid gland.

**Table 3. Median comparison of SSIM and correlation coefficient for *Db2* wavelet with decomposition level 3 and various noise levels and datasets.**

|  | Corr. coeff | | | SSIM | | |
|---|---|---|---|---|---|---|
|  | **Gaussian** | **S&P** | **Speckle** | **Gaussian** | **S&P** | **Speckle** |
| Abdominal | 0.932 | 0.914 | 0.825 | 0.401 | 0.737 | 0.606 |
| Carotids | 0.927 | 0.915 | 0.813 | 0.420 | 0.762 | 0.693 |
| Neck | 0.912 | 0.924 | 0.846 | 0.557 | 0.752 | 0.693 |
| MSK | 0.962 | 0.981 | 0.873 | 0.550 | 0.758 | 0.679 |
| Mean | 0.933 | 0.921 | 0.839 | 0.482 | 0.750 | 0.668 |

**Table 4. The results of computational complexity of Daubechies wavelets for various type of noise and used datasets.** Time is in seconds.

| | Decomp. lvl 3 | | | Decomp. lvl 4 | | |
|---|---|---|---|---|---|---|
| | Gaussian | S&P | Speckle | Gaussian | S&P | Speckle |
| Abdominal | 2395 | 2494 | 2307 | 2253 | 2289 | 2415 |
| Carotids | 2215 | 2313 | 2115 | 1985 | 2004 | 2089 |
| Neck | 1140 | 1192 | 1103 | 1106 | 1134 | 1145 |
| MSK | 1773 | 1834 | 1835 | 1872 | 1826 | 1802 |

**Table 5. The results of computational complexity of Symlet wavelets for various type of noise and used datasets.** Time is in seconds.

| | Decomp. lvl 3 | | | Decomp. lvl 4 | | |
|---|---|---|---|---|---|---|
| | Gaussian | S&P | Speckle | Gaussian | S&P | Speckle |
| Abdominal | 11251 | 11273 | 11513 | 11391 | 12473 | 14447 |
| Carotids | 9294 | 9329 | 11643 | 9878 | 9824 | 12234 |
| Neck | 5202 | 5239 | 5253 | 5247 | 5267 | 5330 |
| MSK | 11698 | 8476 | 8346 | 8119 | 8159 | 8210 |

Fig 3 4) shows the musculoskeletal structures of the knee joint. The bone surface with an anechogenic acoustic shadow is markedly echogenic. Cartilage is anechogenic. Ligaments and tendons appear as slightly hyperechogenic structures with a regular linear structure. Muscle and subcutaneous adipose tissue are normally of low echogenicity with a regular architecture of alternating hyperechogenic linear structures. We can evaluate articular effusions that are predominantly anechogenic in nature.

Subjective evaluation of filtration efficiency is shown in Fig 13. Selected anatomical structures of interest are labelled as follows: vena portae—black indicator, bile duct and hepatic artery branches with white and green indicators, vena cava inferior—blue indicator, and finally ligamentum venosum with orange indicator. We can see that significant anatomical areas of interest are undiagnosable when the image is degraded by higher level of speckle noise, see Fig 13 b). In particular, the area of the vena portae (black indicator) and the area of the hepatic arteries and bile duct (white and green indicators). According to Fig 13 c) we can see a significant noise suppression and a partial improvement in image diagnosability.

## Discussion

As demonstrated by the obtained results, the performance of image denoising depends strongly on the setting of the filtering method used. Of the selected objective criteria, the PSNR parameter seems to be the most relevant as it is the most often used image quality metric. However, it is not recommended to use a single metric to evaluate one's results. Each of the metrics offers different point of view on the image quality and is associated with certain weaknesses and differs on its degree of sensitivity to image degradations. For example, PSNR is more sensitive to additive Gaussian noise than the SSIM as demonstrated in [51]. From representation perspective, SSIM, Q-index, and correlation coefficient are easier to work with since they are normalized, whereas MSE and PSNR are only showing absolute errors [52]. Moreover, SSIM was designed to take into account luminance, contrast, and structure, similarly as the human visual system [53]. This makes it theoretically the most suitable parameter to be used for this task, however, in practice, it does not have to relate to a radiologist's perception of diagnostic image quality [54]. Therefore, we used more parameters to assess the

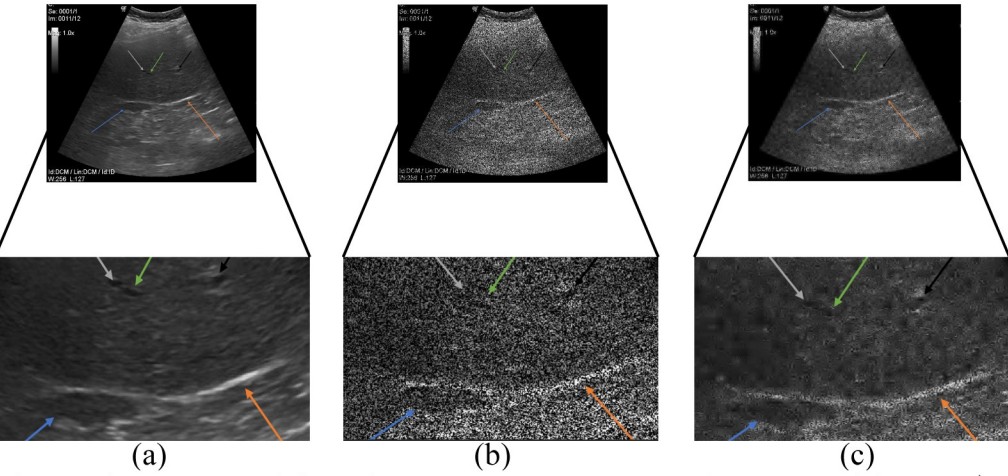

**Fig 13. A example of filtration result by *db2*, level of decomposition 4.** a) native image, b) noisy image (speckle noise ($\mu = 0$, $\sigma^2 = 1$) and c) filtered MSK image.

filtration quality. A system achieving the best results according to all (or most) of the metrics was considered as the most effective one.

Our extensive analysis shows that Daubechies performed best at lower levels of decomposition, even when comparing computational complexity, see Tables 4 and 5. Interestingly, for Gaussian noise, the *Db5* seems to be most effective for all tested databases. The mother wavelet *Db2* slightly outperformed other tested wavelets for Speckle noise and *Db30* for Salt and Pepper noise, see Table 6.

Thus, the insights relative to the effect of the system settings presented in this study may provide a useful guideline for researchers, software developers, and clinical professionals to obtain high quality images from both the technical and clinical points of view. The strengths and contributions of the proposed study can be summarized as follows:

1. *Dataset uniqueness and size*—the tests were carried out on real data; the dataset consisted of in total of 292 ultrasound images and included several areas of interest, such as abdomen (100 images), neck (40 images), carotid (80 images) or musculoskeletal (72 images). Moreover, to extend the dataset and to simulate variety of noise types and levels, we carried out artificial image deterioration using Gaussian, salt and pepper and multiplicative speckle noise. We applied 20 different levels with various parameters. Thus, the final dataset consisted of 17,520 images (292x3x20). Other studies on this topic were conducted either on solely synthetic data such as [48, 55] or on a limited real dataset. For example, in [56] they used objective metrics on synthetic data and on real data, they used only 3 US images and then evaluated the performance subjectively. Subsequently, they used breast ultrasound image database containing 109 cases for experiment to demonstrate the improvement of classification by using the tested denoising algorithms. Further, the authors in [33] introduced a new wavelet type *Usi* and tested it on 110 images from various areas, however, it was tailored for the speckle noise and thus was not tested on any other noise.

2. *Evaluation criteria*—Besides conventional metrics, such as MSE, PSNR and SSIM, this study includes the insights from the clinical experts, who evaluated the filtration results from the clinical point of view. This is important since the objective image quality assessment can only cover limited factors influencing the image quality, such as brightness or

**Table 6. Mean comparison of PSNR for Daubechies family with decomposition level 3 and various noise levels and datasets.**

| Mother Wavelet | Database | Gaussian | Salt&Pepper | Speckle |
|---|---|---|---|---|
| Db2 | MSK | 19.68 ± 4.28 | 25.74 ± 3.30 | **23.79 ± 1.85** |
| | Abdominal | 18.74 ± 3.47 | 25.14 ± 3.31 | **21.32 ± 2.19** |
| | Carotids | 18.76 ± 3.49 | 24.86 ± 3.32 | 21.51 ± 2.12 |
| | Neck | 18.53 ± 3.33 | 25.51 ± 3.24 | **22.39 ± 1.52** |
| Db5 | MSK | **19.83 ± 4.42** | 26.13 ± 3.12 | 23.70 ± 1.94 |
| | Abdominal | **18.75 ± 3.47** | 25.60 ± 3.08 | 20.97 ± 2.27 |
| | Carotids | **18.83 ± 3.54** | 25.20 ± 3.16 | 21.43 ± 2.16 |
| | Neck | **18.58 ± 3.36** | 26.00 ± 3.02 | 22.38 ± 1.56 |
| Db10 | MSK | 19.81 ± 4.28 | 27.95 ± 2.18 | 23.65 ± 1.93 |
| | Abdominal | 18.71 ± 3.44 | 27.21 ± 2.28 | 20.72 ± 2.31 |
| | Carotids | 18.73 ± 3.50 | 26.69 ± 2.37 | 21.45 ± 2.16 |
| | Neck | 18.56 ± 3.34 | 27.51 ± 2.30 | 22.36 ± 1.55 |
| Db15 | MSK | 19.76 ± 4.37 | 29.10 ± 1.74 | 23.64 ± 1.92 |
| | Abdominal | 18.69 ± 3.42 | 28.07 ± 1.97 | 20.59 ± 2.34 |
| | Carotids | 18.77 ± 3.49 | 27.62 ± 2.00 | 21.51 ± 2.15 |
| | Neck | 18.52 ± 3.32 | 28.21 ±2.09 | 22.34 ± 1.52 |
| Db20 | MSK | 19.75 ± 4.36 | 29.75 ± 1.52 | 23.53 ± 1.94 |
| | Abdominal | 18.67 ± 3.41 | 28.46 ± 1.84 | 20.61 ± 2.34 |
| | Carotids | 18.75 ± 3.47 | 28.07 ± 1.82 | 21.59 ± 2.14 |
| | Neck | 18.51 ± 3.31 | 28.49 ± 2.03 | 22.35 ± 1.49 |
| Db22 | MSK | 19.79 ± 4.39 | 30.65 ± 1.27 | 23.59 ± 1.96 |
| | Abdominal | 18.67 ± 3.40 | 28.85 ± 1.88 | 20.62 ± 2.34 |
| | Carotids | 18.74 ± 3.46 | 28.67 ± 1.73 | 21.62 ± 2.13 |
| | Neck | 18.50 ± 3.30 | 28.66 ± 2.13 | 22.35 ± 1.48 |
| Db25 | MSK | 19.82 ± 4.41 | 31.02 ± 1.19 | 23.54 ± 1.97 |
| | Abdominal | 18.66 ± 3.40 | 28.95 ± 1.93 | 20.66 ± 2.33 |
| | Carotids | 18.73 ± 3.46 | 28.85 ± 1.74 | 21.66 ± 2.12 |
| | Neck | 18.50 ± 3.30 | **28.68 ± 2.20** | 22.36 ± 1.46 |
| Db30 | MSK | 19.80 ± 4.39 | **31.48 ± 1.21** | 23.59 ± 1.94 |
| | Abdominal | 18.66 ± 3.40 | **29.01 ± 2.10** | 20.73 ± 2.32 |
| | Carotids | 18.72 ± 3.45 | **29.05 ± 1.90** | **21.74 ± 2.11** |
| | Neck | 18.49 ± 3.29 | 28.61 ± 2.36 | 22.38 ± 1.44 |

contrast. However, even though the results of the objective evaluations are outstanding, the clinical requirements are often not met. As mentioned in [57], filtering allows a better separation of classes between asymptomatic and symptomatic subjects. Both the perception and interpretation of medical visual information are critical in clinical practice. However, medical images are not self-explanatory and, therefore, need to be interpreted by the medical experts, whose quality of experience and thus their decision may be impacted by the image distortions or unsatisfactory filtration.

3. *Extension of current research outcomes*—Our work extends the work of Adamo et al. [48] and provides new extensive investigations in the selection of the wavelet base, as the method was tested on in total of 292 real ultrasound images. It also offers a unique clinical-based evaluation of ultrasound images to ensure that the proposed method preserves all the clinically important information. In our work, orthogonal wavelets were selected because they

provide more precise and consistent results as already mentioned in [48], where both biorthogonal and orthogonal filters were tested.

4. *New findings*—this study demonstrates that the denoising system using Daubechies and lower-level decomposition improves ultrasound noise-canceling procedure in terms of objective and subjective evaluation. When applied on real ultrasound images, we could observe only a slight deviation (±2.2%) among different types of noise and thus this system setting can be considered as robust and effective for this application.

Despite all the above-mentioned strengths of the proposed study, there are also some limitations and possibilities for future research. When analyzing the wavelet settings for various noise, we do not obtain a comparison with other images with the same conditions. In this context, it would be worth studying a simultaneous system response in the form of a spatial 2D distribution of evaluation parameters for various Wavelet settings. Besides the 1D trends for dynamical noise influence, we would receive an immediate simultaneous response to any number of Wavelet settings for specific noise settings. Such a tool should have a strong potential for a comparative evaluation of the Wavelet settings for specific conditions in ultrasound image processing.

The absence of extensive subjective evaluation may be also considered as a limitation of this study. The blind questionnaire for image quality assessment by experts could not be conducted as the artificially degraded dataset counted a total of 17,520 images, which were subsequently filtered by various selection of mother wavelet functions leading to creating tens of thousands of images. However, using the results obtained in this study, a more robust setting can be selected and used for the evaluation tests. Moreover, the tests should be carried out on vast amount of data from clinical practice, where different image degradations may occur, as mentioned in the introduction. Then, the questionnaire survey will be carried out to evaluate each filtrated image.

However, at this stage, we only wanted to provide clinical insight on selected noise and dataset. This is because this article only focuses on the preprocessing stage, where the aim is to reduce the significant amount of noise present in the images. In the subsequent stage, the image enhancement needs to be carried out to obtain the clinically important features and or to be able to apply the segmentation methods and so on. Therefore, the subjective evaluation by experts is crucial in these subsequent phases rather than in the preprocessing. This will be a subject of the future research.

## Conclusion

This study provides an extensive analysis and a quantitative evaluation of various wavelet denoising systems, their settings for different types of noise, and other effects influencing the quality of the resulting ultrasound image. The extensive analysis on in total of 17,520 images (dataset created from 292 real ultrasound images) shows that both the filtration system setting and the image content, namely type of noise and selected dataset, play a crucial role in the quality of the filtration. The performance of the tested methods was assessed by conventional objective metrics (correlation coefficient MSE, PSNR, SSIM, and Q-index). For selected filtration outcomes, we also provided clinical insights from clinical experts. The results showed that Daubechies at lower-level of decomposition achieved the best results. Namely, *Db2*, *Db5*, and *Db30*.

Moreover, the obtained results also indicate that it is not possible to determine the universal type of wavelet for variant types of noise. The choice of the most effective wavelet should be

tailored for a specific purpose since it depends on the type of tested ultrasound images, especially the selected area of interest, device used, and type of noise present.

## Author Contributions

**Conceptualization:** Dominik Vilimek.

**Data curation:** Dominik Vilimek.

**Formal analysis:** Dominik Vilimek, Jan Kubicek, Milos Golian, Rene Jaros, Radana Kahankova, Pavla Hanzlikova, Daniel Barvik, Alice Krestanova.

**Funding acquisition:** Marek Penhaker, Ondrej Prokop.

**Investigation:** Dominik Vilimek.

**Methodology:** Dominik Vilimek.

**Project administration:** Martin Cerny.

**Resources:** Dominik Vilimek, Milos Golian, Pavla Hanzlikova, Marek Buzga.

**Software:** Dominik Vilimek.

**Supervision:** Jan Kubicek.

**Validation:** Radana Kahankova.

**Visualization:** Dominik Vilimek, Rene Jaros, Radana Kahankova.

**Writing – original draft:** Dominik Vilimek, Jan Kubicek.

**Writing – review & editing:** Jan Kubicek, Rene Jaros, Radana Kahankova, Marek Buzga.

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
