## [Decision Letter · Decision Letter 0]

18 Nov 2021

PONE-D-21-28525Comparative Analysis of Wavelet Transform Filtering Systems for Noise Reduction in Ultrasound ImagesPLOS ONE

Dear Dr. Vilimek,

Thank you for submitting your manuscript to PLOS ONE. After careful consideration, we feel that it has merit but does not fully meet PLOS ONE’s publication criteria as it currently stands. Therefore, we invite you to submit a revised version of the manuscript that addresses the points raised during the review process.

We look forward to receiving your revised manuscript.

Kind regards,

Yiming Tang, Ph.D.

Academic Editor

PLOS ONE

Journal Requirements:

The authors have declared that no competing interests exist.

We note that one or more of the authors are employed by a commercial company: MEDIN

Reviewers' comments:

Reviewer's Responses to Questions

**Comments to the Author**

1. Is the manuscript technically sound, and do the data support the conclusions?

Reviewer #1: No

2. Has the statistical analysis been performed appropriately and rigorously? 

Reviewer #1: No

3. Have the authors made all data underlying the findings in their manuscript fully available?

Reviewer #1: Yes

4. Is the manuscript presented in an intelligible fashion and written in standard English?

Reviewer #1: Yes

5. Review Comments to the Author

Reviewer #1: 1) p. 7 “Interestingly, with increasing noise level, the efficiency of Db5 slightly decreases while in case of Db25 the efficiency increases, see Figure 4b 4c and 4d. However, SSIM metric shows that Db10 and Db25 achieved better results. In the case of higher level of decomposition similar trend could be found.” These conclusions are not supported by the figures. Differences between the curves are seen when they do not exist. For example in 4c and 4d a difference between the curves are described while in each of the figures the curves are all grouped together and are visually indistinguishable. Decrease or increase of efficiency cannot be stated with certainty these modifications are very little noticeable. These remarks hold for other figures also.

2) Depending on the criterion (correlation coefficient, MSE, PSNR, Q-index, or SSIM)., the performances of the wavelets chosen are markedly different. What is the most relevant criterion in relation to the final objective, which is the clinical interpretation of the images? this should be analyzed with the assistance of radiologists.

3) The conclusion « When applied on real ultrasound images, we could observe only a slight deviation ( 2:2%) among different types of noise and thus this system setting can be considered as robust and effective for this application.” Should be better justified and indicate how the deviation of 2.2% is obtained

4) In the abstract, the authors clam that “this study includes subjective evaluations from clinical experts, who assessed the filtration results” and “…to obtain high quality images from both the technical and clinical points of view”. These assertion on the contribution to clinical aspect are not supported by this study.

5) For the evaluation of clinical analysis performed by radiologists, only the case of MSK image is considered, other types of images are ignored. This is an important limitation to the interest of the paper since aiding reliable radiological analysis is the end goal of this work. Moreover in the filtered image (c) of figure 13, we can observe that many artifacts are enhanced and can be confusing for the visual detection of veins and it is impossible to determine the right veins if we only have the filtered image. A relevant analysis should be made by offering several radiologists to blindly analyze the images and compare their conclusions with the results obtained from the initial images.

Minor remarks:

- Line 119/120 « see example is represents in Figure 2”, please correct.

- Lines 146-148. Please correct “soft-threshing” (probably soft-thresholding) and “perceive” (probably “preserve”)

- Caption Figure 6 is incomplete. Please correct.

6. PLOS authors have the option to publish the peer review history of their article (what does this mean?). If published, this will include your full peer review and any attached files.

Reviewer #1: No

---

## [Author Response · Author response to Decision Letter 0]

15 Feb 2022

Dear Editor,

Thank you for you feedback to our manuscript.

Below, we are uploading point-by-point response to the comments of all reviewers. All changes are clearly highlighted with red in the revised manuscript.

Best regards,

Dominik Vilimek et al.

1) p. 7 “Interestingly, with increasing noise level, the efficiency of Db5 slightly decreases while in case of Db25 the efficiency increases, see Figure 4b 4c and 4d. However, SSIM metric shows that Db10 and Db25 achieved better results. In the case of higher level of decomposition similar trend could be found.” 

These conclusions are not supported by the figures. Differences between the curves are seen when they do not exist. For example in 4c and 4d a difference between the curves are described while in each of the figures the curves are all grouped together and are visually indistinguishable. Decrease or increase of efficiency cannot be stated with certainty these modifications are very little noticeable. These remarks hold for other figures also.

Authors’ answer: Indeed, the figures were not described properly. This whole part will be corrected.

Authors’ action: We rewrote following sections: Elimination of Gaussian Noise, Elimination of Salt and Pepper Noise, Elimination of Speckle Noise.

2) Depending on the criterion (correlation coefficient, MSE, PSNR, Q-index, or SSIM), the performances of the wavelets chosen are markedly different. 

What is the most relevant criterion in relation to the final objective, which is the clinical interpretation of the images? this should be analyzed with the assistance of radiologists.

Authors’ answer: Of the selected objective criteria, the PSNR parameter seems to be the most relevant as it is the most often used image quality metric. However, it is not recommended to use a single metric to evaluate one’s results. Each of the metrics offers different point of view on the image quality and is associated with certain weaknesses and differ on their degree of sensitivity to image degradations. For example, PSNR is more sensitive to additive Gaussian noise than the SSIM as demonstrated in [1]. 

From representation perspective, SSIM, Q-index and correlation coefficient are easier to work with since they are normalized, whereas MSE and PSNR are not and are only showing absolute errors [2]. Moreover, SSIM was designed to take into account luminance, contrast, and structure, similarly as the human visual system [3]. This makes it theoretically the most suitable parameter to be used for this task, however, in practice, it does not have to relate to a radiologist's perception of diagnostic image quality [4]. Therefore, we used more parameters to assess the quality of the filtration and the system achieving the best results according to all (or most) of the metrics was considered as the most effective one.

It is important to mention that post-processing of ultrasound images is not a standard in current clinical practice. However, image processing can be of great benefit, especially in practices without top ultrasound devices or in cases where an increase in transmitted energy is undesirable. Moreover, current research shows that using the SSIM metric as a loss function (or component) for image enhancement using Deep Learning algorithms produces better results. Therefore, an objective evaluation based on only one parameter is unfavorable in view of future developments. Loss Function selection is a key component of today's ML and DL algorithms. Where metrics like SSIM, MSE, PSNR can be used to estimate image quality loss. Filtration is only one part of image enhancement techniques, where wavelet transformation still plays a major role [5]. 

Authors’ action: We added the above mention to discussion section. 

3) The conclusion When applied on real ultrasound images, we could observe only a slight deviation ( 2:2%) among different types of noise and thus this system setting can be considered as robust and effective for this application.” 

Should be better justified and indicate how the deviation of 2.2% is obtained

Authors’ answer: Thank you for pointing this out. We have revised the conclusion and added Table 6 to discussion.

Authors’ action: We modified the conclusion and demonstrated the obtained results for Daubechies family in Table 6.

4) In the abstract, the authors clam that “this study includes subjective evaluations from clinical experts, who assessed the filtration results” and “…to obtain high quality images from both the technical and clinical points of view”. 

These assertion on the contribution to clinical aspect are not supported by this study.

Authors’ answer: Thank you for your comment. This contribution was introduced in incorrect manner and will be corrected.

Authors’ action: We modified the abstract:

Wavelet transform (WT) is a commonly used method for noise suppression and feature extraction from biomedical images. The selection of WT system settings significantly affects the efficiency of denoising procedure. This comparative study analyzed the efficacy of the proposed WT system on real 292 ultrasound images from several areas of interest. The study investigates the performance of the system for different scaling functions of two basic wavelet bases, Daubechies and Symlets, and their efficiency on images artificially corrupted by three kinds of noise. To evaluate our extensive analysis, we used objective metrics, namely structural similarity index (SSIM), correlation coefficient, mean squared error (MSE), peak signal-to-noise ratio (PSNR) and universal image quality index (Q-index). Moreover, this study includes clinical insights on selected filtration outcomes provided by clinical experts. The results show that the efficiency of the filtration strongly depends on the specific wavelet system setting, type of ultrasound data, and the noise present. The findings presented may provide a useful guideline for researchers, software developers, and clinical professionals to obtain high quality images.

5) For the evaluation of clinical analysis performed by radiologists, only the case of MSK image is considered, other types of images are ignored. This is an important limitation to the interest of the paper since aiding reliable radiological analysis is the end goal of this work. Moreover, in the filtered image (c) of figure 13, we can observe that many artifacts are enhanced and can be confusing for the visual detection of veins and it is impossible to determine the right veins if we only have the filtered image. A relevant analysis should be made by offering several radiologists to blindly analyze the images and compare their conclusions with the results obtained from the initial images.

Authors’ answer: The goal of this work was to test the influence of various factors (such as wavelet system setting, or type and level of noise) on the filtration quality. This is necessary step to optimally set the system and ensure the quality of the outcomes. We discussed the evaluation aspect with radiologists. Objective evaluation in our case seems to be ideal for comparing individual settings and choosing the optimal type of mother wave, which is the aim of this article. For comparison, 292 images were created, which were artificially degraded by three variant noises of 20 levels, resulting in 17,520 images. These were subsequently filtered by systems different WT settings (16 different mother waves with two different decomposition level settings) leading to creating resulting tens of thousands of images. Based on these analyzes, we have a better idea of the robustness of each type of wave.

Due to the enormous number of images, it is beyond the scope of the article to provide subjective analysis of all of the images. This will be the goal of future work, where we aim to carry out the tests with the most suitable system setting on vast amount of data from clinical practice, where different image degradations may occur, as mentioned in the introduction. Then we aim to focus on the specific settings and impact of the clinical assessment for each image and conduct a questionnaire survey to determine the final conclusions. However, at this stage, we only wanted to provide clinical insight on selected noise and dataset. Nevertheless, we realize that we may have introduced this fact in incorrect manner thus we will made modifications in the manuscript.

Authors’ action:

1) We modified the abstract:

Wavelet transform (WT) is a commonly used method for noise suppression and feature extraction from biomedical images. The selection of WT system settings significantly affects the efficiency of denoising procedure. This comparative study analyzed the efficacy of the proposed WT system on real 292 ultrasound images from several areas of interest. The study investigates the performance of the system for different scaling functions of two basic wavelet bases, Daubechies and Symlets, and their efficiency on images artificially corrupted by three kinds of noise. To evaluate our extensive analysis, we used objective metrics, namely structural similarity index (SSIM), correlation coefficient, mean squared error (MSE), peak signal-to-noise ratio (PSNR) and universal image quality index (Q-index). Moreover, this study includes clinical insights on selected filtration outcomes provided by clinical experts. The results show that the efficiency of the filtration strongly depends on the specific wavelet system setting, type of ultrasound data, and the noise present. The findings presented may provide a useful guideline for researchers, software developers, and clinical professionals to obtain high quality images.

2) We added this aspect as limitation of this study in the discussion:

The absence of extensive subjective evaluation may be considered as a limitation of this study. The blind questionnaire for image quality assessment by experts could not be conducted as the artificially degraded dataset counted a total of 17,520 images, which were subsequently filtered by various selection of mother wavelet functions leading to creating tens of thousands of images. However, using the results obtained in this study, a more robust setting can be selected and used for the evaluation tests. Moreover, the tests should be carried out on vast amount of data from clinical practice, where different image degradations may occur, as mentioned in the introduction. Then, the questionnaire survey will be carried out to evaluate each filtrated image. However, at this stage, we only wanted to provide clinical insight on selected noise and dataset. 

3) We modified the conclusion:

This study provides an extensive analysis and a quantitative evaluation of various wavelet denoising systems, their settings for different types of noise, and other effects influencing the quality of the resulting ultrasound image. The extensive analysis on in total of 17,520 images (dataset created from 292 real ultrasound images) shows that both the filtration system setting and the image content, namely type of noise and selected dataset, play a crucial role in the quality of the filtration. The performance of the tested methods was assessed by conventional objective metrics (correlation coefficient MSE, PSNR, SSIM, and Q-index). For selected filtration outcomes, we also provided clinical insights from clinical experts. The results showed that Daubechies at lower-level of decomposition achieved the best results. Namely, Db2, Db5, and Db30. 

However, the obtained results also indicate that it is not possible to determine the universal type of wavelet for variant types of noise. The choice of the most effective wavelet should be tailored for a specific purpose since it depends on the type of tested ultrasound images, especially the selected area of interest, device used, and type of noise present.

[1] Hore, A., & Ziou, D. (2010, August). Image quality metrics: PSNR vs. SSIM. In 2010 20th international conference on pattern recognition (pp. 2366-2369). IEEE.

[2] Sara, U., Akter, M., & Uddin, M. S. (2019). Image quality assessment through FSIM, SSIM, MSE and PSNR—a comparative study. Journal of Computer and Communications, 7(3), 8-18.

[3] Setiadi, D. R. I. M. (2021). PSNR vs SSIM: imperceptibility quality assessment for image steganography. Multimedia Tools and Applications, 80, 8423-8444.

[4] A. Mason et al., "Comparison of Objective Image Quality Metrics to Expert Radiologists’ Scoring of Diagnostic Quality of MR Images," in IEEE Transactions on Medical Imaging, vol. 39, no. 4, pp. 1064-1072, April 2020, doi: 10.1109/TMI.2019.2930338

[5] Liu, JW., Zuo, FL., Guo, YX. et al. Research on improved wavelet convolutional wavelet neural networks. Appl Intell 51, 4106–4126 (2021). https://doi.org/10.1007/s10489-020-02015-5

---

## [Decision Letter · Decision Letter 1]

29 Apr 2022

PONE-D-21-28525R1Comparative Analysis of Wavelet Transform Filtering Systems for Noise Reduction in Ultrasound ImagesPLOS ONE

Dear Dr. Vilimek,

Thank you for submitting your manuscript to PLOS ONE. After careful consideration, we feel that it has merit but does not fully meet PLOS ONE’s publication criteria as it currently stands. Therefore, we invite you to submit a revised version of the manuscript that addresses the points raised during the review process.

We look forward to receiving your revised manuscript.

Kind regards,

Yiming Tang, Ph.D.

Academic Editor

PLOS ONE

Reviewers' comments:

Reviewer's Responses to Questions

**Comments to the Author**

1. If the authors have adequately addressed your comments raised in a previous round of review and you feel that this manuscript is now acceptable for publication, you may indicate that here to bypass the “Comments to the Author” section, enter your conflict of interest statement in the “Confidential to Editor” section, and submit your "Accept" recommendation.

Reviewer #1: All comments have been addressed

Reviewer #2: (No Response)

2. Is the manuscript technically sound, and do the data support the conclusions?

Reviewer #1: Yes

Reviewer #2: No

3. Has the statistical analysis been performed appropriately and rigorously? 

Reviewer #1: No

Reviewer #2: I Don't Know

4. Have the authors made all data underlying the findings in their manuscript fully available?

Reviewer #1: Yes

Reviewer #2: No

5. Is the manuscript presented in an intelligible fashion and written in standard English?

Reviewer #1: Yes

Reviewer #2: No

6. Review Comments to the Author

Reviewer #1: The manuscript is now more in line with the data and results obtained. For this purpose, the authors have modified the comments of several figures to better reflect what they actually represent.

The authors also moderated their assertions concerning the radiological validity of the methods used. They recognize that the absence of extensive subjective evaluation may be considered as a limitation of this study. They argue that blind questionnaire for image quality assessment by experts could not be conducted as the artificially degraded dataset counted a total of 17,520 images, which were subsequently filtered by various selection of mother wavelet functions leading to creating tens of thousands of images. This argument is not entirely acceptable since the evaluation can be carried out on a well-chosen subset of the totality of the images. However, the limitation mentioned by the authors is sufficient in itself.

Taking into account the substantial modifications made to the manuscript, it can now be accepted for publication

Reviewer #2: In this paper, the authors present a comparison of wavelets transforms dedicated to denoising ultrasound images that were artificially corrupted with several types of noise. They estabish the most suitable transform depends on the type of noise but also the type of region.

The comparison is interesting and the numerous metrics and experiments could make the study very solid. However, two crucial things are missing in the paper :

1 - the figures ! The 13 figures that should be present in the manuscript do not show on the PDF, as well as several parts such as section numbers. There was very probably an error of rendering. Either way this hinders a complete review as these figures are essential to understand the methodology and the results.

2. A clear clinical interest. The authors work on denoising images that were artificially made noisy. This noise is added mathematically, not linked to any real aspect of the acquisition. It is understandable that this allows to have as groundtruth the original image. But the original image is the one we would like to denoise. How is denoising added noise relevant to the current problems radiologists encounter in their clinical practice ? This should be made clearer.

7. PLOS authors have the option to publish the peer review history of their article (what does this mean?). If published, this will include your full peer review and any attached files.

Reviewer #1: No

Reviewer #2: No

---

## [Author Response · Author response to Decision Letter 1]

17 May 2022

Dear Editor,

Thank you for you feedback to our manuscript. We would like to take this opportunity and thank the reviewers for their tremendous work on reviewing our manuscript. The changes greatly improved the quality and readability of our manuscript.

Below, we are uploading point-by-point response to the comments of all reviewers. The changes of the second round of the reviews were highlighted in blue, the first round changes were kept red.

Best regards,

Dominik Vilimek et al.

Reviewer 1

Comment 1: The manuscript is now more in line with the data and results obtained. For this purpose, the authors have modified the comments of several figures to better reflect what they actually represent.

The authors also moderated their assertions concerning the radiological validity of the methods used. They recognize that the absence of extensive subjective evaluation may be considered as a limitation of this study. They argue that blind questionnaire for image quality assessment by experts could not be conducted as the artificially degraded dataset counted a total of 17,520 images, which were subsequently filtered by various selection of mother wavelet functions leading to creating tens of thousands of images. This argument is not entirely acceptable since the evaluation can be carried out on a well-chosen subset of the totality of the images. However, the limitation mentioned by the authors is sufficient in itself.

Taking into account the substantial modifications made to the manuscript, it can now be accepted for publication

Authors’ answer: Thank you for all your valuable comments and time you spent reviewing our manuscript, it significantly helped in enhancing its quality. We incorporate your further comments in our future work to remove the remaining limitations.

 

Reviewer 2

In this paper, the authors present a comparison of wavelets transforms dedicated to denoising ultrasound images that were artificially corrupted with several types of noise. They estabish the most suitable transform depends on the type of noise but also the type of region.

The comparison is interesting and the numerous metrics and experiments could make the study very solid. However, two crucial things are missing in the paper:

Comment 1: The figures ! The 13 figures that should be present in the manuscript do not show on the PDF, as well as several parts such as section numbers. There was very probably an error of rendering. Either way this hinders a complete review as these figures are essential to understand the methodology and the results.

Authors’ answer: We agree that this is confusing, however, this is the way Plos One journal’s template is, please see:

The figures were uploaded separately according to the guidelines and can be downloaded from the PLOS One repository (as .rar file). We are sorry about this confusion, which was not caused by us. 

Authors’ action: To make it easier to follow the manuscript, we are also uploading the version that has the figures incorporated in the text, this version will be uploaded as a supporting material, please see document entitled “manuscript_with_figures.pdf”.

Comment 2: A clear clinical interest. The authors work on denoising images that were artificially made noisy. This noise is added mathematically, not linked to any real aspect of the acquisition. It is understandable that this allows to have as groundtruth the original image. But the original image is the one we would like to denoise. How is denoising added noise relevant to the current problems radiologists encounter in their clinical practice ? This should be made clearer.

Authors’ answer: We agree that it may seem odd to use artificial noise on the images. There are different ways to test the data, as described in the Discussion. The most common approach is to use solely synthetic data or phantoms. The second is by evaluating the data visually – either by simply plotting the data before and after filtration or by using experts who would evaluate their diagnostic quality (e.g. using questionaries).

However, in our opinion, the approach used in our study is the only one suitable to test the highest amount of noise types and filter settings possible, while using the real (base) images and being able to objectively evaluate the outputs. When we would only use the real images corrupted by the noise, we would only be able to assess it subjectively – using the experts’ opinions, where we would be limited by the number of images being scored and accuracy of such results (inter/intra observer reliability). We tried to make these experiments as realistic as possible: 1) by using high amount of real images acquired; 2) by using the types of noise that are most common in clinical practice (Speckle noise, Gaussian noise, and Salt and Pepper noise); and 3) we also added a part where the clinicians had a chance to comment on the quality of the images (see section Analysis performed by radiologists (lines 315 – 360) and Fig. 13). 

As a result, we were able to denoise the US image while preserving the diagnostically important features. However, an extensive subjective evaluation is missing in our study which we consider as a major limitation of our study, as we mentioned in discussion. The blind questionnaire for image quality assessment by experts could not be conducted due to large number of images in the dataset (17,520 subsequently filtered by various filters, i.e. tens of thousands of images to be evaluated). However, using the results obtained in this study, a more robust setting can be selected and used for the evaluation tests of a smaller dataset size.

Finally, this article only focuses on the preprocessing stage, where the aim is to reduce the significant amount of noise present in the images. In the subsequent stage, the image enhancement needs to be carried out to obtain the clinically important features and or to be able to apply the segmentation methods and so on. Therefore, the subjective evaluation by experts is crucial in these subsequent phases rather then in the pre-processing. This will be a subject of the future research. 

We tried to enhance these facts in the manuscript.

Authors’ action: We highlighted the above-mentioned aspects in the text as follows:

1) Introduction:

Page 2, line 25-31

There are three noise types typical for ultrasound (US) imaging: 1) Speckle noise is the most characteristic and prevalent one, it can affect important image details and may influence the intensity parameters, such as contrast; 2) Gaussian noise is caused by sensor or electronic circuit noise; 3) Salt and pepper noise occurs due to sudden changes in an image, such as memory cell failure, synchronization error during digitalization or improper function of the sensor cells.

Page 2, line 32-42

The presence of the above-mentioned noise types generally leads to degradation of visual US image quality. Thus, it is important to test the efficacy of the denoising procedure on various types of noise. In this paper we are focusing on the image preprocessing, where we often employ so-called image enhancement methods for US image noise-canceling. The preprocessing methods are aimed at noise removal and include mathematical algorithms, which can at least partially reduce the noise from US images. Image preprocessing has a substantial importance for further steps of image processing, including identification and extraction of objects of interest from ultrasound images. Images corrupted with noise or artifacts deteriorate the pixels distribution, thus decreasing performance of the image segmentation techniques such as regional and semantic segmentations.

Page 3, line 95-102

To test the highest amount of noise types and filter settings possible, we corrupted created dataset with noise generators to simulate various image impairments occurring in US images (Speckle, Gaussian, and Salt and Pepper noise). This way, we used the real US images (serving as ground truth) and were able to objectively evaluate the outputs. In contrast to subjective evaluation by experts, which is associated with certain limitations, such as inter/extra observer disagreement.

2) Discussion:

Page 13, line 397-405

Other studies on this topic were conducted either on solely synthetic data such as [48, 55] or on a limited real dataset. For example, in [56] they used objective metrics on synthetic data and on real data, they used only 3 US images and then evaluated the performance subjectively. Subsequently, they used breast ultrasound image database containing 109 cases for experiment to demonstrate the improvement of classification by using the tested denoising algorithms. Further, the authors in [33] introduced a new wavelet type Usi and tested on 110 images from various areas, however, it was tailored for the speckle noise and thus was not tested on any other noise.

Page 15, line 450-456

However, at this stage, we only wanted to provide clinical insight on selected noise and dataset. This is because this article only focuses on the preprocessing stage, where the aim is to reduce the significant amount of noise present in the images. In the subsequent stage, the image enhancement needs to be carried out to obtain the clinically important features and or to be able to apply the segmentation methods and so on. Therefore, the subjective evaluation by experts is crucial in these subsequent phases rather than in the preprocessing. This will be a subject of the future research.

---

## [Decision Letter · Decision Letter 2]

17 Jun 2022

Comparative Analysis of Wavelet Transform Filtering Systems for Noise Reduction in Ultrasound Images

PONE-D-21-28525R2

Dear Dr. Vilimek,

We’re pleased to inform you that your manuscript has been judged scientifically suitable for publication and will be formally accepted for publication once it meets all outstanding technical requirements.

Kind regards,

Yiming Tang, Ph.D.

Academic Editor

PLOS ONE

Additional Editor Comments (optional):

Reviewers' comments:

Reviewer's Responses to Questions

**Comments to the Author**

1. If the authors have adequately addressed your comments raised in a previous round of review and you feel that this manuscript is now acceptable for publication, you may indicate that here to bypass the “Comments to the Author” section, enter your conflict of interest statement in the “Confidential to Editor” section, and submit your "Accept" recommendation.

Reviewer #1: All comments have been addressed

Reviewer #2: All comments have been addressed

2. Is the manuscript technically sound, and do the data support the conclusions?

Reviewer #1: Yes

Reviewer #2: Yes

3. Has the statistical analysis been performed appropriately and rigorously? 

Reviewer #1: No

Reviewer #2: Yes

4. Have the authors made all data underlying the findings in their manuscript fully available?

Reviewer #1: Yes

Reviewer #2: Yes

5. Is the manuscript presented in an intelligible fashion and written in standard English?

Reviewer #1: Yes

Reviewer #2: Yes

6. Review Comments to the Author

Reviewer #1: (No Response)

Reviewer #2: (No Response)

7. PLOS authors have the option to publish the peer review history of their article (what does this mean?). If published, this will include your full peer review and any attached files.

Reviewer #1: No

Reviewer #2: No

---

## [Editor Report · Acceptance letter]

24 Jun 2022

PONE-D-21-28525R2 

Comparative Analysis of Wavelet Transform Filtering Systems for Noise Reduction in Ultrasound Images 

Dear Dr. Vilimek:

I'm pleased to inform you that your manuscript has been deemed suitable for publication in PLOS ONE. Congratulations! Your manuscript is now with our production department. 

Kind regards, 

on behalf of

Professor Yiming Tang 

Academic Editor

PLOS ONE